# FoundTS: Comprehensive and Unified Benchmarking of Foundation Models for Time Series Forecasting

## Abstract

Time Series Forecasting (TSF) is key functionality in numerous fields, including in finance, weather services, and energy management. While TSF methods are emerging these days, many of them require domain-specific data collection and model training and struggle with poor generalization performance on new domains. Foundation models aim to overcome this limitation. Pre-trained on large-scale language or time series data, they exhibit promising inferencing capabilities in new or unseen data. This has spurred a surge in new TSF foundation models. We propose a new benchmark, `FoundTS`, to enable thorough and fair evaluation and comparison of such models. `FoundTS` covers a variety of TSF foundation models, including those based on large language models and those pretrained on time series. Next, `FoundTS` supports different forecasting strategies, including zero-shot, few-shot, and full-shot, thereby facilitating more thorough evaluations. Finally, `FoundTS` offers a pipeline that standardizes evaluation processes such as dataset splitting, loading, normalization, and few-shot sampling, thereby facilitating fair evaluations. Building on this, we report on an extensive evaluation of TSF foundation models on a broad range of datasets from diverse domains and with different statistical characteristics. Specifically, we identify pros and cons and inherent limitations of existing foundation models, and we identify directions for future model design. We make our code and datasets available at https://anonymous.4open.science/r/FoundTS-C2B0.

## 1 Introduction

Time Series Forecasting (TSF) is core functionality in a multitude of applications, including in finance, weather services, and energy management (Wu et al., 2024; Pan et al., 2023; Wan et al., 2022; Qin et al., 2023). Given historical observations, predicting future states is valuable for decision making and taking appropriate actions. Consequently, TSF is a very active research field, as evidenced by the continued emergence of prediction models. However, most existing TSF models require training on specific datasets in preparation for performing inference on corresponding datasets. Such models (called *specific models* in this paper to be distinguished from *foundation models*) do not generalize well and experience suboptimal performance when applied to new or unseen data (Shi et al., 2024; Huang et al., 2023; Nie et al., 2022; Lin et al., 2024). Efforts to address challenges such as these have led to a recent surge in the development of foundation models for TSF (Liang et al., 2024; Woo et al., 2024; Liu et al., 2024b; Zhou et al., 2024).

While foundation models encompassing diverse architectures and training paradigms continue to appear, our understanding of their strengths and limitations remains limited. Existing research on understanding such models has focused primarily on the qualitative analysis and categorization of foundation models for TSF (Liang et al., 2024; Jin et al., 2023b; 2024). For instance, (Jin et al., 2024) propose a potential framework for LLM-based time series analysis, highlighting key opportunities and challenges for future research and advocating for increased interdisciplinary collaboration and exploration in this promising field. Similarly, (Liang et al., 2024) adopt a methodology-centered classification approach, outlining critical elements of time series foundation models, such as model architectures, pre-training techniques, and adaptation strategies. However, these studies often lack a quantitative evaluation of foundation models, which is crucial for assessing and comparing per-

formance. Quantitative analysis not only provides a clearer understanding of model performance; it also enables researchers to make informed decisions about model selection and improvements.

Further, proposals for foundation models often adopt different experimental setups, making it difficult to compare meaningfully the performance of different foundation models based only on the existing body of performance studies. As shown in Table 1, few-shot learning studies employ different types of sampling.

Table 1: Comparison of experimental settings.

| Setting | Zero-shot sampling type | Sampling ratio | Lookback length |
|---|---|---|---|
| Timer | Uniform window sampling | 1–75% | 672 |
| UniTS | Uniform window sampling | 5%,15%,20% | 96 |
| TimeLLM | Front-end window sampling | 5%,10% | 512 |
| S2IP-LLM | Front-end window sampling | 5%,10% | 512 |

Some use uniform window sampling (Liu et al., 2024b; Gao et al., 2024), while others use front-end window sampling (Jin et al., 2023a; Pan et al., 2024). Additionally, studies employ different lookback lengths and sampling ratios.

Robust and thorough benchmarks enable researchers to evaluate new models more rigorously, which is crucial for advancing the state-of-the-art (Tan et al., 2020; Qiao et al., 2024). Most time series forecasting benchmarks target performance evaluation of specific models, while evaluations of foundation models are relatively scarce—see Table 2. The only benchmark targeting foundation models (Zhang et al., 2023) has two notable limitations: 1) the types of foundation models considered is not sufficiently comprehensive, as LLM-based models are ignored; 2) the evaluation strategies supported are too limited to reflect fully the performance of foundation models, as few-shot settings are not considered.

Motivated by these observations, we present `FoundTS`, a benchmark designed to facilitate fair and comprehensive empirical evaluation and comparison of time series foundation models. First, to assess the performance of models thoroughly, `FoundTS` includes datasets spanning different domains and with diverse characteristics. Second, `FoundTS` covers a variety of time series foundation models, including LLM-based models and pre-trained models. This is in addition to state-of-the-art specific models that are included to enable comparison with the foundation models. (3) Third, `FoundTS` supports multiple evaluation strategies, including zero-shot, few-shot, and full-shot approaches, and employs a variety of evaluation metrics to evaluate model performance more thoroughly. Fourth, to ensure fair comparisons, `FoundTS` provides an experimental setup that standardizes processes such as dataset splitting, loading, normalization, and few-shot sampling. These properties combine to enable a fair and complete pipeline, ensuring thorough evaluations with findings that are comparable. `FoundTS` thus enables comprehensive evaluation of time series foundation models, providing reliable insight into their characteristics and pros and cons. In addition, we identify inherent limitations in current foundation models and offer directions for future model design. In summary, we make the following main contributions:

- **Diversified models and datasets:** `FoundTS` covers state-of-the-art time series foundation models, including LLM-based and pre-trained time series models. Additionally, it features comprehensive datasets that encompass a wide range of domains and characteristics.

- **Comprehensive and fair evaluation strategies and pipelines:** `FoundTS` integrates zero-shot, few-shot, and full-shot approaches, facilitating improved assessment of model performance. Further, it provides a unified experimental setup that standardizes dataset splitting, loading, and few-shot sampling, thereby facilitating fair comparisons of models.

- **In-depth quantitative analysis and insights:** Employing `FoundTS`, we report on extensive experiments that cover different time series foundation models. This way, we identify pros and cons of the time series models covered and offer insights for us in future model design and optimization.

## 2 RELATED WORK

### 2.1 TIME SERIES FORECASTING

TSF models can be categorized as **specific models** and **foundation models**. The former typically require training on specific datasets and inferencing on corresponding datasets. Three categories of such models exist. **Statistical learning models** (Box & Jenkins, 1968; Hyndman et al., 2008), though theoretically robust, struggle to capture nonlinear trends, thus limiting their predictive accu-

Table 2: Comparison between `FoundTS` and other time series forecasting benchmarks.

| Time Series Forecasting Benchmark | Evaluated Models | | | Evaluation Strategies | | |
|---|---|---|---|---|---|---|
| | LLM-based models | TS pre-trained models | Specific models | Zero-shot | Few-shot | Full-shot |
| M3 (Makridakis & Hibon, 2000) | × | × | √ | × | × | √ |
| M4 (Makridakis et al., 2018) | × | × | √ | × | × | √ |
| LTSF-Liner (Zeng et al., 2023) | × | × | √ | × | × | √ |
| BasicTS (Liang et al., 2022) | × | × | √ | × | × | √ |
| BasicTS+ (Shao et al., 2023) | × | × | √ | × | × | √ |
| Monash (Godahewa et al., 2021) | × | × | √ | × | × | √ |
| Libra (Bauer et al., 2021) | × | × | √ | × | × | √ |
| ProbTS (Zhang et al., 2023) | × | √ | √ | √ | × | √ |
| TSLib (Wang et al., 2024d) | × | × | √ | × | × | √ |
| TFB (Qiu et al., 2024) | × | × | √ | × | × | √ |
| FoundTS (ours) | √ | √ | √ | √ | √ | √ |

racy. **Machine learning models** (Chen & Guestrin, 2016; Friedman, 2001) are better at capturing non-linear relationships and complex patterns, but often require manual feature engineering and model design. **Deep learning models** (Zhong et al., 2023; Dai et al., 2024; Wen et al., 2023; Wang et al., 2024a; Lin et al., 2023; Zhou et al., 2021) leverage the representation learning capabilities of deep neural networks on rich datasets, often outperforming the other two categories of techniques at predictive accuracy. However, all these models are limited by their strong coupling of training and inferencing data. They may not perform well on new or unseen data.

Foundation models for time series forecasting (Liang et al., 2024) can be divided into two categories: **LLM-based models** and **time series pre-trained models**. Both exhibit outstanding zero-shot and few-shot prediction capabilities on unseen time series datasets. LLM-based models (Zhou et al., 2024), with their vast language understanding and context processing cap abilities, are capable of high-quality forecasts when faced with unseen data. Time series pre-trained models (Woo et al., 2024; Liu et al., 2024b), through pre-training on large time series datasets, exhibit generalization capabilities, enabling them to perform forecasting with limited training data.

## 2.2 TIME SERIES FORECASTING BENCHMARKS

Several benchmarks have recently been proposed for TSF—see Table 2. However, their inherent limitations make comprehensive and fair comparison of foundation models and specific models for time series forecasting out of reach. First, most benchmarks target only specific models and ignore time series foundation models. As shown in Table 2, the only exception is ProbTS (Zhang et al., 2023), which, however, only cover time series pre-trained models, not LLM-based models. With foundation models now offering impressive features like zero-shot prediction, there is a need for fair and thorugh comparisons with specific models, considering the challenges they pose, such as high computational costs.

Second, current benchmarks do not support diverse evaluation strategies. Most time series forecasting benchmarks disregard emerging features like zero-shot and few-shot prediction, focusing instead on full-shot scenarios. While ProbTS (Zhang et al., 2023) supports zero-shot prediction, it does not support few-shot prediction, which enables models to leverage small amounts of relevant data to fine-tune their performance. This capability not only enhances accuracy but also increases a model's flexibility at adapting to new tasks, making it more effective in dynamic environments and across diverse environments. Furthermore, the absence of standardized sampling methods for few-shot prediction compromises fair model comparison. There is a need for a more inclusive and fair evaluation pipeline.

`FoundTS` is designed to be a more reliable, comprehensive, and user-friendly benchmark, featuring a wider range of TSF models and evaluation strategies. In addition, it offers a unified experimental setup, ensuring consistent model evaluations within a robust pipeline.

## 3 FOUNDTS

To facilitate the evaluation and comparison of forecasting foundation models, we propose `FoundTS`, a unified benchmark for time series forecasting foundation models. Figure 1 shows its three core modules: Data, Model, and Evaluation. The data module includes time series datasets from different domains and with diverse characteristics, providing comprehensive data support for

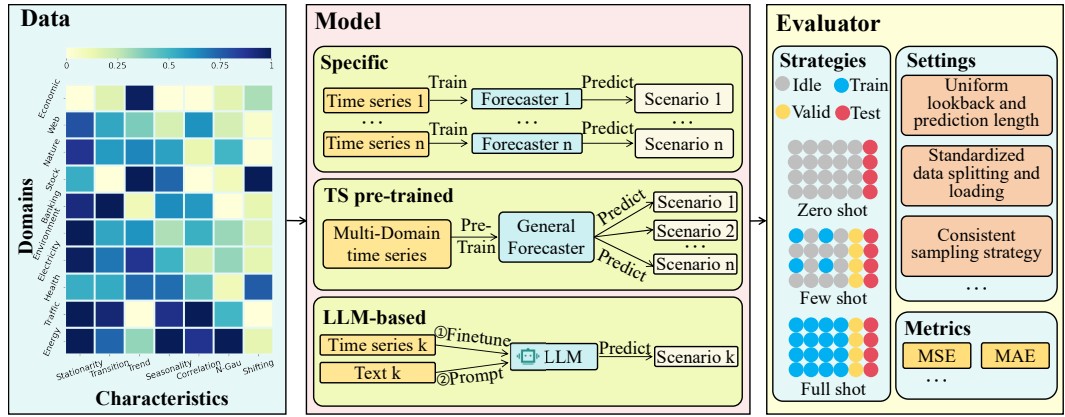

Figure 1: The `FoundTS` architecture with three core modules: Data, Model, and Evaluation.

downstream time series forecasting. The model module includes time series foundation models, including LLM-based models pretrained with large-scale text and time series pre-trained models pretrained with multi-domain time series, along with specific models. The evaluation module offers a scalable pipeline and standardized evaluation environment with comprehensive strategies and consistent settings, ensuring fair comparisons of models and facilitating reliable results.

## 3.1 DATA

High-quality and diverse time series data enable comprehensive evaluation of model performance, facilitating the selection of models that are most suitable for specific forecasting scenarios. The data offers broad coverage of domains as well as statistical characteristics, to more comprehensively compare the prediction and generalization performance of models.

(1) **Domains**: We include datasets from ten domains, including *stock* (NASDAQ (Feng et al., 2019)), *health* (ILI (Wu et al., 2021)), *energy* (Solar (Lai et al., 2018)), *electricity* (ETT (Zhou et al., 2021) and Electricity (Trindade, 2015)), *environment* (Weather (Wu et al., 2021)), *traffic* (Traffic (Wu et al., 2021)), *nature* (ZafNoo (Poyatos et al., 2020)), *banking* (NN5 (Taieb et al., 2012)), *web* (Wike2000 (Gasthaus et al., 2019)), and *economics* (Exchange (Lai et al., 2018)), for evaluation.

(2) **Characteristics**: We consider a range of important time series characteristics, including seasonality, trend, stationarity, transition, shifting, correlation, and non-Gaussianity (Qiu et al., 2024; Zhang et al., 2023). *Seasonality* refers to repeating patterns or cycles at regular intervals. *Trend* indicates overall movements in a time series. *Stationarity* reflects the statistical properties of a time series, such as mean and variance, which do not change over time. *Transition* represents sudden or gradual shifts in a time series. *Shifting* refers to changes in the level or timing of the data and includes vertical and horizontal shifts. *Correlation* represents the relationship or dependence among different channels. *Non-Gaussianity (N-Gau)* represents deviations from normal distribution, often exhibiting skewness or kurtosis. The formula used to calculate these characteristics can be found in Appendix B. The "Data" part of Figure 1 shows data domains with varying characteristic distributions. This facilitates comprehensive evaluation of prediction accuracy and generalization cap abilities under varying data characteristics. More details of the datasets are included in Appendix A.1.

## 3.2 MODELS

### 3.2.1 TIME SERIES FOUNDATION MODELS

**LLM-based models**: LLMs-based methods leverage the strong representational capacity and sequential modeling capability of LLMs to capture complex patterns for time series modeling. To more comprehensively evaluate the foundation models, we incorporate existing LLMs-based methods into `FoundTS`, focusing primarily on parameter-efficient fine-tuning and prompting. 1) Parameter-efficient fine-tuning methods: GPT4TS (Zhou et al., 2024), S²IPLLM (Pan et al., 2024) selectively adjust specific parameters such as positional encoding and layer normalization of LLMs, enabling the model to quickly adapt to time series while retaining most of pre-trained knowledge. 2) Prompt-

ing methods, such as UniTime (Liu et al., 2024a) and Time-LLM (Jin et al., 2023a), focus on designing prompts, such as learnable prompts, prompt pools, and domain-specific instructions to activate time series knowledge in LLMs.

**Time-series pre-trained models**: Pre-training on multi-domain time series data has gained significant attention in recent years. We incorporate time series pre-trained models into `FoundTS`, categorizing them into four types based on the pre-training approach: reconstruction, autoregressive, direct prediction, and hybrid training. 1) Reconstruction methods: MOIRAI Woo et al. (2024), UniTS Gao et al. (2024), Moment Goswami et al. (2024) restore the features of time series data, enabling them to extract valuable information in an unsupervised manner. This type of method mainly adopts the encoder architecture. 2) Autoregressive methods: TimesFM Das et al. (2023), Timer Liu et al. (2024b), employ next token prediction to learn time series representation. This type of method mainly adopts the decoder architecture. 3) Direct prediction methods: TTM Ekambaram et al. (2024), unify the training process between pre-training and downstream tasks, allowing models to exhibit strong adaptability when transitioning to downstream forecasting tasks. 4) Hybrid pre-training methods: ROSE Wang et al. (2024c), combines the strengths of both reconstruction and direct prediction to learn generalized time series representations.

### 3.2.2 TIME SERIES SPECIFIC MODELS

Time series specific models typically require training on specific datasets and perform inference on the corresponding datasets. To better showcase the capabilities of time series foundation models, we select several SOTA specific models for comparison. We include: 1) CNN-based models: TimesNet (Wu et al., 2022), which treat time series as sequences of vectors and leverage CNNs to capture temporal dependencies. 2) Transformer-based models: FEDformer (Zhou et al., 2022), iTransformer (Liu et al., 2023), and PatchTST (Nie et al., 2022), which are capable of capturing more complex temporal dynamics, leading to significantly improved forecasting performance. 3) MLP-based models: FITS (Xu et al., 2024), TimeMixer (Wang et al., 2024b), and DLinear (Zeng et al., 2023), with their simple architecture and relatively few parameters, have demonstrated strong forecasting accuracy as well.

### 3.3 EVALUATION

To ensure a fair and comprehensive evaluation of the performance of various time series forecasting models, we standardize the evaluation in three key areas: strategies, settings, and metrics.

### 3.3.1 STRATEGIES

Considering that current benchmarks typically adopt a single evaluation approach, focusing only on zero-shot or full-shot scenarios, this limits the ability to comprehensively assess prediction performance. We propose a more comprehensive quantitative evaluation that offers researchers a broader understanding under different conditions, including zero-shot, few-shot, and full-shot. As shown in Figure 1, we divide the downstream evaluation data into train, validation, and test data.

(1) The zero-shot evaluation only uses the test data to evaluate the generalization ability of foundation models to new datasets, assessing whether the model has truly learned general knowledge from vast amounts of pre-training data.

(2) The few-shot evaluation only utilizes a subset of training data and full validation data for fine-tuning, reflecting the prediction performance in low-data learning scenarios. This approach assesses whether models can effectively generalize and reason with minimal data support.

(3) The full-shot evaluation utilizes full train data and validation data for fine-tuning. It evaluates the performance when utilizing all available data, revealing its upper-bound performance.

### 3.3.2 SETTINGS AND METRICS

Different evaluation settings can cause significant discrepancies in model performance, leading to unfair comparisons of their actual capabilities. To address this, we standardize the settings, including lookback and prediction length, data splits, and sampling strategy.

(1) **Uniform lookback and prediction lengths**: The lookback length determines the amount of historical information the model receives, and different lengths lead to varying prediction results. Following the common practices, we consider four prediction lengths: 24, 36, 48, and 60, for NASDAQ, NN5, ILI, and Wike2000; and we use another four prediction lengths 96, 192, 336, and 720, for all other datasets which have longer lengths. The lookback lengths underwent testing with lengths 36 and 104 for NASDAQ, NN5, ILI, and Wike2000, and 96, 336, and 512 for all other datasets. For each prediction length, we report the best performance across different lookback lengths.

(2) **Standardized data splitting and loading**: We standardize the division of the training, validation and test datasets, as well as the partitioning of each time series sample for all models. To ensure that different models use a consistent test length, we do not apply the "Drop Last" operation during testing (Qiu et al., 2024).

(3) **Consistent sampling strategies**: We integrate various sampling strategies, including random sampling, uniform sampling, front-end sampling, and back-end sampling. Besides, we support both window and point sampling format—see Table 3. The results demonstrate that different sampling types significantly impact model performance, even leading to substantial performance gaps. This indicates that data sampling types play a crucial role in few-shot learning, and standardized experimental setups are essential for fairly evaluating the actual performance of foundation models. By default, we consistently use 5% uniform window sampling across all models for assessment and reporting to ensure a fair comparison. Our pipeline supports the seamless transition to other strategies.

Table 3: Comparison of different sampling types.

| Strategy | ETTm1 Format | Timer MAE | Timer MSE | UniTS MAE | UniTS MSE | TTM MAE | TTM MSE |
|---|---|---|---|---|---|---|---|
| Random | Window sample[1] | **0.351** | **0.299** | 0.477 | 0.568 | 0.386 | 0.361 |
| Uniform | Window sample | **0.345** | **0.288** | 0.449 | 0.474 | 0.368 | 0.330 |
| Front-end | Window sample | 0.425 | 0.456 | 0.511 | 0.682 | **0.401** | **0.387** |
| | Point sample[2] | **0.410** | 0.416 | 0.515 | 0.684 | 0.414 | **0.407** |
| Back-end | Window sample | 0.365 | 0.313 | 0.442 | 0.466 | 0.374 | 0.339 |
| | Point sample | 0.374 | 0.322 | 0.452 | 0.489 | 0.375 | 0.341 |

[1] Window sample refers to first dividing the dataset into windows (lookback length + prediction length), and then selecting a specified proportion of these window samples.
[2] Point sample refers to directly extracting a specified proportion of data points from the dataset.

We incorporate a variety of metrics for evaluation, including Mean Absolute Error (MAE) and Mean Squared Error (MSE), among others. Different metrics offer a multifaceted evaluation of model performance, each providing unique insights from different perspectives.

## 4 EXPERIMENTS

### 4.1 BENCHMARKING RESULTS

#### 4.1.1 ZERO-SHOT EVALUATION

Specific models require training on data of each specific scenario, and most LLM-based models need fine-tuning of either the LLM backbones or some additional components to adapt to downstream datasets. Thus, in zero-short evaluation, we focus on time series pre-trained models that are capable of performing zero-shot forecasting. We present the zero-shot performance of these time series pre-trained models in Table 4. The main findings are as follows: 1) It is evident that no single model consistently outperforms others across all datasets. Such variation may be due to differences in model architectures, pre-training datasets and tasks, and model sizes among current pre-trained models. 2) Compared with the few-shot results of specific models in Table 5, pre-trained models demonstrate superior performance on smaller datasets, such as ETTh1 and Exchange. On larger datasets like Electricity and Weather, pre-trained models perform on par with specific models. Compared with the full-shot results of specific models in Table 6, pre-trained models outperform

Table 4: Pre-trained model results in the **zero-shot setting**. The results are the average MSE of all prediction lengths. The complete zero-shot results of MAE and MSE can be found in Appendix D.1.

| Model | ETTh1 | ETTh2 | ETTm1 | ETTm2 | Electricity | Traffic | Solar | Weather | Exchange | ZafNoo | ILI | NASDAQ | NN5 | Wike2000 |
|---|---|---|---|---|---|---|---|---|---|---|---|---|---|---|
| TimesFM | 0.479 | 0.400 | **0.435** | 0.347 | **0.154** | **0.370** | **0.500** | **0.226** | 0.389 | 0.631 | **3.025** | 1.034 | **0.780** | **475.582** |
| Timer | 0.451 | 0.365 | 0.544 | **0.298** | 0.257 | 0.612 | 0.660 | 0.259 | 0.393 | 0.557 | 3.523 | **0.890** | 1.259 | 605.069 |
| UniTS | 0.414 | 0.374 | 0.761 | 0.335 | 0.198 | 0.497 | 0.845 | 0.725 | 0.423 | 0.716 | 4.364 | 1.195 | 1.292 | 612.973 |
| TTM | 0.403 | 0.349 | 0.779 | 0.338 | 0.219 | 0.611 | 0.775 | 0.252 | 0.343 | 0.535 | 4.595 | 1.667 | 1.333 | 502.890 |
| MOIRAI | 0.431 | 0.360 | 0.561 | 0.337 | 0.241 | - | 0.785 | 0.321 | 0.386 | **0.522** | 3.407 | 1.045 | 0.796 | - |
| ROSE | **0.401** | **0.346** | 0.525 | 0.299 | 0.234 | 0.588 | 0.517 | 0.265 | 0.618 | 0.544 | 4.606 | 1.131 | 1.336 | 649.503 |

MOIRAI flattens all channels into a single dimension for patching, thus limiting its use when dealing with datasets with many channels. MOIRAI fails to work on Traffic (862 channels) and Wike2000 (2000 channels), which is shown with –.

Table 5: Model results in the **5% few-shot setting**. The results are the average MSE of all prediction lengths. The complete few-shot results of MAE and MSE can be found in Appendix D.2.

| | Model | ETTh1 | ETTh2 | ETTm1 | ETTm2 | Electricity | Traffic | Solar | Weather | Exchange | ZafNoo |
|---|---|---|---|---|---|---|---|---|---|---|---|
| | TimesFM | 0.459 | 0.357 | 0.785 | 0.350 | 0.285 | 0.718 | 0.559 | 0.343 | 0.470 | 0.702 |
| | Timer | 0.406 | 0.349 | 0.351 | 0.268 | 0.175 | 0.420 | **0.202** | **0.231** | 0.349 | 0.509 |
| TS | UniTS | 0.426 | 0.369 | 0.551 | 0.302 | 0.170 | 0.426 | 0.232 | 0.244 | 0.413 | 0.587 |
| Pre-trained | TTM | 0.400 | 0.345 | 0.761 | 0.303 | 0.220 | 0.630 | 0.883 | 0.255 | **0.335** | 0.523 |
| Models | Moment | 0.468 | 0.369 | 0.374 | 0.270 | 0.246 | 0.776 | 0.530 | 0.249 | 0.451 | 0.546 |
| | MOIRAI | 0.452 | 0.397 | 0.521 | 0.385 | 0.228 | - | 5.259 | 0.293 | 0.400 | 0.921 |
| | ROSE | **0.399** | 0.337 | **0.349** | **0.253** | 0.174 | 0.417 | 0.206 | 0.242 | 0.513 | 0.536 |
| | GPT4TS | 0.466 | 0.372 | 0.386 | 0.272 | 0.206 | 0.424 | 0.251 | 0.252 | 0.412 | 0.564 |
| LLM-based | S²IPLLM | 0.683 | 0.396 | 0.408 | 0.303 | - | - | 0.314 | 0.246 | 0.407 | 0.758 |
| Models | UniTime | 0.800 | 0.421 | 0.409 | 0.274 | 0.202 | 0.433 | 0.218 | 0.294 | 0.458 | 0.815 |
| | Time-LLM | 0.674 | 0.369 | 0.373 | 0.270 | - | - | - | 0.238 | 0.407 | 0.600 |
| | PatchTST | 0.462 | 0.386 | 0.374 | 0.280 | 0.179 | 0.432 | 0.268 | 0.245 | 0.379 | 0.544 |
| | DLinear | 0.459 | 0.500 | 0.381 | 0.296 | **0.168** | 0.440 | 0.247 | 0.250 | 0.361 | **0.506** |
| Specific | FITS | 0.408 | **0.335** | 0.357 | 0.254 | 0.176 | 0.436 | 0.232 | 0.244 | 0.349 | 0.527 |
| Models | iTransformer | 0.572 | 0.411 | 0.415 | 0.287 | 0.227 | 0.479 | 0.251 | 0.269 | 0.416 | 0.699 |
| | FEDformer | 0.557 | 0.477 | 0.666 | 0.418 | 0.303 | 0.836 | 0.711 | 0.364 | 0.643 | 0.683 |
| | TimesNet | 0.855 | 0.449 | 0.483 | 0.323 | 0.263 | 0.871 | 0.378 | 0.284 | 0.438 | 0.612 |
| | TimeMixer | 0.627 | 0.365 | 0.376 | 0.273 | 0.202 | 0.475 | 0.225 | 0.235 | 0.368 | 0.533 |

The maximum training duration was constrained to a maximum of 5 hours. Models that exceeded this threshold are represented with –.

specific models on datasets such as ETTh1, Exchange, and NASDAQ. However, they still fall short on datasets like Weather and ILI. These results suggest that while pre-trained models show promising zero-shot capabilities, they have not yet fully surpassed the need for data-intensive training.

### 4.1.2 FEW-SHOT EVALUATION

We assess time series pre-trained models, LLM-based models, and specific models with a 5% few-shot setting, and the results are reported in Table 5. The main findings are as follows: 1) Time series pre-trained models generally outperform both LLM-based and specific models, with 7 out of 10 datasets showing a lead. This advantage likely stems from their ability to capture fundamental temporal patterns during pre-training, which enables quicker adaptation to downstream datasets. This highlights the time series pre-trained model's superiority in data-efficiency or under conditions of data scarcity, demonstrating that they can still maintain high performance when dealing with limited data. 2) Most pre-trained models show improved few-shot results compared to zero-shot performance, particularly on large datasets like Solar and Electricity. Notably, Timer and ROSE excel in the few-shot setting. 3) Some LLM-based models, such as GPT4TS, outperform certain specific models, but the majority of LLM-based models perform worse than SOTA specific models. This disparity may be attributed to the cross-modal information in texts compared to time series, which renders them less effective for time series tasks. 4) A few specific models, such as FITS and DLinear, achieve strong performance in some datasets, potentially because their smaller parameter sizes allow for faster fitting of simple time-series information. This indicates that research on few-shot learning may not only focus on foundation models but also on some efficient small models.

### 4.1.3 FULL-SHOT EVALUATION

Since full-shot training on some foundation models may take substantially long time, which violates the original intention of the foundation models, we only select several representative foundation models that are more efficient in training in the full-shot setting. As shown in Table 6: 1) Since the specific models are trained using all available training data, its performance is somewhat superior to that of the foundation models, with 4 out of 7 datasets showing a lead. This indicates that the foundation models still has room for improvement in full-shot scenarios. It also suggests that in the future, we can enhance the overall performance of the foundation models by further optimizing its structure and training strategies. 2) pre-trained models and specific models outperform LLM-based models in prediction accuracy; however, the performance gap between LLM-based models and pre-trained models is narrowing. 3) Compared to the few-shot results, some pre-trained models, such as Timer, show a decline in performance in the full-shot setting. This suggests that pre-trained models are more suitable in few-shot and zero-shot scenarios. 4) LLM-based models perform better in the full-shot setting compared to their performance in the few-shot setting, likely because the increase in training data helps unlock time-series-related knowledge embedded in the LLMs.

Table 6: Model results in the **full-shot setting**. The results are the average MSE of all prediction lengths. The complete full-shot results of MAE and MSE can be found in Appendix D.3.

| | Model | ETTh1 | Weather | Exchange | ZafNoo | ILI | NASDAQ |
|---|---|---|---|---|---|---|---|
| TS Pre-trained Models | Timer | 0.500 | 0.296 | 0.418 | 0.561 | 5.808 | 1.248 |
| | UniTS | 0.453 | **0.224** | 0.609 | 0.529 | 2.845 | 1.175 |
| | TTM | **0.397** | 0.247 | **0.349** | 0.512 | 4.409 | 1.382 |
| LLM-based Models | GPT4TS | 0.420 | 0.231 | 0.475 | 0.514 | 3.764 | 1.177 |
| | UniTime | 0.483 | 0.230 | 0.437 | 0.520 | 3.299 | 1.099 |
| Specific Models | PatchTST | 0.411 | 0.225 | 0.352 | 0.512 | **1.770** | 0.977 |
| | DLinear | 0.420 | 0.239 | 0.349 | 0.496 | 2.185 | 1.504 |
| | FITS | 0.408 | 0.244 | 0.349 | 0.527 | 2.051 | 1.070 |
| | iTransformer | 0.439 | 0.233 | 0.360 | 0.535 | 1.801 | 0.987 |
| | FEDFormer | 0.432 | 0.307 | 0.483 | 0.578 | 2.185 | **0.976** |
| | TimesNet | 0.459 | 0.261 | 0.421 | 0.537 | 2.174 | 0.998 |
| | TimeMixer | 0.436 | 0.226 | 0.383 | 0.518 | 1.821 | 1.142 |

Table 7: The results of loading pre-trained parameters (denotes as "p") and random initialization (denotes as "w/o p") for foundation models in 5% few-shot setting.

| | Weather | | | | ETTh2 | | | |
|---|---|---|---|---|---|---|---|---|
| Model | p | | w/o p | | P | | w/o p | |
| | MAE | MSE | MAE | MSE | MAE | MSE | MAE | MSE |
| TimesFM | 0.449 | 0.436 | 0.340 | 0.274 | 0.366 | 0.315 | 0.717 | 0.891 |
| Timer | 0.212 | 0.161 | 0.212 | 0.161 | 0.348 | 0.290 | 0.404 | 0.371 |
| UniTS | 0.276 | 0.204 | 0.233 | 0.182 | 0.381 | 0.327 | 0.412 | 0.397 |
| TTM | 0.205 | 0.153 | 0.199 | 0.151 | 0.342 | 0.283 | 0.394 | 0.361 |
| Moment | 0.239 | 0.182 | 0.240 | 0.182 | 0.377 | 0.328 | 0.442 | 0.440 |
| MOIRAI | 0.266 | 0.215 | 0.333 | 0.284 | 0.350 | 0.300 | 0.343 | 0.356 |
| ROSE | 0.205 | 0.159 | 0.225 | 0.179 | 0.332 | 0.272 | 0.354 | 0.309 |
| GPT4TS | 0.244 | 0.187 | 0.222 | 0.169 | 0.377 | 0.322 | 0.368 | 0.314 |
| $S^2$IPLLM | 0.228 | 0.171 | 0.227 | 0.175 | 0.415 | 0.366 | 0.392 | 0.345 |
| Time-LLM | 0.220 | 0.167 | 0.219 | 0.165 | 0.418 | 0.372 | 0.369 | 0.314 |
| UniTime | 0.239 | 0.184 | 0.211 | 0.158 | 0.421 | 0.375 | 0.397 | 0.353 |

## 4.2 ANALYSIS ON DIFFERENT FOUNDATION MODELS

### 4.2.1 CHANNEL INDEPENDENCE VS. CHANNEL DEPENDENCE

In multivariate datasets, variables are often referred to as channels. To explore the impact of channel dependency in multivariate time series, we compare MOIRAI, Moment, iTransformer, and Times-Net across ten datasets with varying degrees of correlations, ranging from weak to strong. We present the MSE result for forecasting 96 in Figure 2. Our findings show that foundation models that account for channel dependence, such as MOIRAI, typically outperform those that assume channel independence (e.g., Moment) on datasets with strong correlations. However, in some cases, MOIRAI's performance is outperformed by specific models that also consider channel dependence, such as iTransformer and TimesNet. This reflects that MOIRAI's way of handling correlation is not as smart as the specific models. This calls for foundation models that use more appropriate way of modeling correlations.

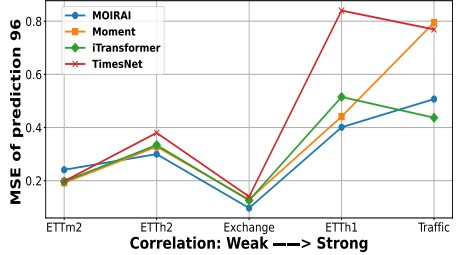

Figure 2: Reports models few-shot performance for varying correlation within datasets.

Figure 3: A comparison of the parameter counts and pre-training dataset sizes of pre-trained models, along with their zero-shot performance.

### 4.2.2 COMPARISON AMONG DIFFERENT ARCHITECTURES

From Figure 3, we observe that the TimesFM model, based on the Transformer architecture and possessing the largest number of parameters, achieves optimal performance. Surprisingly, the model TTM, based on a multi-layer perceptron (MLP) and with the smallest number of parameters, performs just below TimesFM and ROSE, while outperforming many foundation models with significantly larger model parameter sizes. This result prompts us to rethink existing architectures. Specifically, this phenomenon indicates that current architectures do not fully reflect the "scaling law," and existing time series foundation models do not necessarily show a positive correlation between model parameters and performance. Therefore, while TimesFM leads in performance, its increase in parameters is not the only path to performance enhancement. This finding suggests that there is ample room for development in the study of time series foundation models. In future research, we need to delve deeper into model architecture design to find a better trade off between performance and parameter counts. Additionally, innovative network structures, such as hybrid architectures, may provide new insights for improving time series data modeling capabilities.

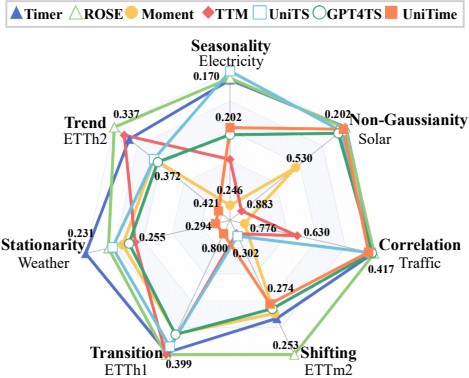

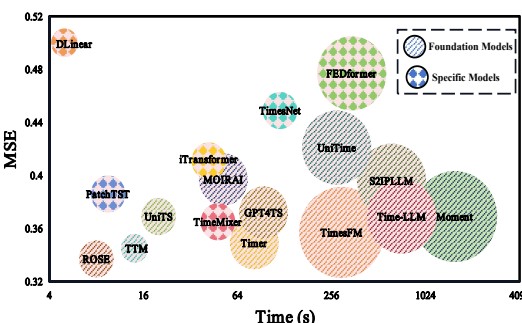

Figure 4: **5% few-shot MSE** for foundation models across seven data characteristics.

Figure 5: Model efficiency. The size of the circle represents the size of each model parameter.

### 4.2.3 PERFORMANCE ON DIFFERENT DATA CHARACTERISTICS

We evaluate the performance of foundation models across different characteristics. We first score the time series datasets with respect to the above seven characteristics. For each characteristic, we select the dataset with the highest score to represent it. We present the 5% few-shot MAE results for the models in Figure 4. Results reveal that no single foundation model excels across all characteristics. Notably, ROSE demonstrates exceptional performance on datasets where the transition is highly pronounced (ETTh1), exhibits significant trends (ETTh2), or experiences severe drift (ETTm2). Meanwhile, Timer achieves optimal performance on datasets with strong correlation (Traffic), pronounced non-gaussianity (Solar), and most stationary (Weather). Similarly, UniTS stands out for its performance on time series with strong seasonality (Electricity). Timer and ROSE show consistent performance across all datasets, without any significantly poor outcomes. In contrast, TTM model falls short on the ETTm2 dataset, while the Moment model struggles on the Traffic dataset.

### 4.2.4 PRETRAIN VS. NO PRETRAIN

To assess the practical benefits of pre-trained knowledge derived from multi-domain time series data and text data for downstream time series prediction tasks, we select the ETTh2 and Weather datasets and conduct 5% few-shot experiments on existing time series pre-trained models and LLM-based models, both with pre-loaded parameters and random initialization. Specifically, for LLM-based models, we randomly initialize the LLM and other parts. The main results are shown in Table 7: 1) all time series pre-trained models with loaded parameters achieve significant improvements compared to random initialization, particularly on the small ETTh2 dataset. The results indicate that pre-trained models significantly benefit from the knowledge obtained from multi-source time series datasets, demonstrating their strong generalization capabilities. 2) In contrast, many LLM-based models exhibit a decline in predictive performance when loading pre-trained parameters, suggesting that the pre-trained knowledge acquired from text data may negatively impact downstream predictions. Therefore, further optimization and redesign of LLM-based models are crucial to effectively leverage their potential. 3) Comparing the performance of random initialized LLM-based models with random initialized pre-trained models, we observe that LLM-based models perform as well as, or even outperform, pre-trained models. This indicates that the LLM-based architecture may be well-suited for time series forecasting tasks and suggests that multi-domain pre-training based on this architecture may achieve good outcomes.

### 4.2.5 MODEL EFFICIENCY ANALYSIS

Model efficiency is a key criterion for assessing whether a foundation model can adapt effectively to new tasks. To assess the relationship between model efficiency and prediction accuracy of various models, we select the ETTh2 dataset and conduct comparisons between the foundation models for 5% few-shot and time series-specific models for full-shot. Specifically, we compare models based on three aspects: run-time, number of model parameters, and prediction accuracy. Run-time includes training time and inference time. As illustrated in Figure 5, most time series pre-trained models

outperform LLM-based models in terms of running time, number of parameters, and prediction accuracy. By comparing with time series specific models, foundation models exhibit varied performance levels. For instance, ROSE and TTM demonstrate superior running efficiency and prediction accuracy compared to most specific models. In contrast, Timer achieves high prediction accuracy but has long runtime. Additionally, models like Moment and $S^2$IPLLM lag behind specific models in both runtime and prediction accuracy. This suggests that researchers can take model efficiency into consideration when designing foundation models.

### 4.3 TAKEAWAYS FROM BENCHMARKING AND ANALYSIS

Based on our benchmarking and analysis with `FoundTS`, we summarize some takeaways considering the following critical questions related to foundation models for time series forecasting.

*Do foundation models outperform specific models?* Current foundation models, especially those time series pre-trained models, exhibit superior zero-shot and few-shot learning abilities compared with specific models, which indicates their advantages in data-insufficient scenarios. However, when sufficient training data are available, foundation models do not consistently outperform specific models with full-shot learning, indicating their limitations in fully utilizing sufficient data.

*Which foundation models are better?* The advantages of different foundation models for time series forecasting depend on diverse aspects of evaluation, and no single model dominates across these aspects. 1) Considering the two types of foundation models, current time series pre-trained models exhibit better overall performance than LLM-based models. 2) Different foundation models show their advantages in dealing with datasets from diverse domains or with diverse characteristics. 3) Large-scale time series pre-trained models such as TimesFM show the best zero-shot performance, but the situation changes given few-shot data for fine-tuning, where models like Timer perform better. 4) The scaling law does not hold strictly in current foundation models for time series forecasting, and some small-sized models such as ROSE and TTM achieve a better balance between precision and efficiency. It calls for benchmarks such as `FoundTS` to provide comprehensive evaluations of foundation models for time series forecasting to answer this question.

*What improvements are needed for foundation models?* 1) More universal capabilities for diverse datasets and scenarios: Considering that no foundation model wins all situations, a meaningful goal is to explore a more universal model for time series forecasting to handle diverse forecasting situations simultaneously. From the comparison between foundation models and specific models, the development of more powerful foundation models should not only consider enhancing data-scarce performance but also increasing the upper bound forecasting performance given more sufficient training data. 2) Better designs for utilizing large-scale pre-training knowledge: Proper training data, architecture, and pre-training strategies need to be investigated to make time series models truly take advantage of the scaling law. Since multivariate appears as a common characteristic in time series, how to embed generalizable correlation modeling in foundation models from large-scale data remains an open problem. For LLM-based models, more in-depth analysis should be made to fully extract and adapt LLM knowledge for time series forecasting. 3) More efficient training and inference: Considering that specific models are easy to train, an efficient foundation model that balances performance and costs is also valuable to make foundation models more practical in real-world applications.

## 5 CONCLUSION

Foundation models for time series forecasting have recently gained significant attention due to their impressive generalization capabilities in zero-shot and few-shot situations, leading to a surge of innovative models. This paper introduces `FoundTS`, the first comprehensive benchmark designed for quantitatively evaluating foundation TSF models. `FoundTS` encompasses a diverse array of state-of-the-art models and includes three experimental scenarios: zero-shot, few-shot, and full-shot. Additionally, it provides a unified pipeline to ensure consistent evaluations. Using `FoundTS`, we thoroughly assess 11 foundation TSF models, revealing their strengths and weaknesses. Furthermore, we highlight the inherent limitations of current models and propose critical directions for future model design. Overall, `FoundTS` and our evaluation offer researchers enhanced tools for developing new foundation TSF models.

# 6 REPRODUCIBILITY

The study meets reproducibility requirements. Specifically, the datasets and the code can be browsed at https://anonymous.4open.science/r/FoundTS-C2B0.

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

# A EXPERIMENTAL DETAILS

## A.1 DATASETS

We use the following 14 multivariate time-series datasets which cover 10 domains for forecasting: ETT Zhou et al. (2021) datasets contain 7 variates collected from two different electric transformers from July 2016 to July 2018. It consists of four subsets, of which ETTh1/ETTh2 are recorded hourly, and ETTm1/ETTm2 are recorded every 15 minutes. Electricity (Trindade, 2015) contains the electricity consumption of 321 customers from July 2016 to July 2019, recorded hourly. Traffic (Wu et al., 2021) contains road occupancy rates measured by 862 sensors on freeways in the San Francisco Bay Area from 2015 to 2016, recorded hourly. Solar (Lai et al., 2018) records solar power generation from 137 PV plants in 2006, every 10 minutes. Weather (Wu et al., 2021) collects 21 meteorological indicators, such as temperature and barometric pressure, for Germany in 2020, recorded every 10 minutes. Exchange (Lai et al., 2018) collects the daily exchange rates of eight countries. ZafNoo (Poyatos et al., 2020) is collected from the Sapflux data project and includes sap flow measurements and environmental variables. ILI (Wu et al., 2021) records indicators of patients data from Centers for Disease Control and Prevention. NASDAQ (Feng et al., 2019) records opening price, closing price, trading volume, lowest price, and highest price. NN5 (Taieb et al., 2012) is from banking and records the daily cash withdrawals from ATMs in the UK. Wike2000 (Gasthaus et al., 2019) is the daily page views of 2000 Wikipedia pages. Table 8 lists statistics of the 14 multivariate time series datasets. Please note that the values for the seven characteristics—seasonality, trend, stationarity, transition, shifting, correlation, and non-Gaussianity—in the table are the results of min-max normalization.

Table 8: The statistics of evaluation datasets

| Dataset | Variables | Timestamps | Split Ratio | Domain | Frequency | Seasonality | Trend | Stationarity | Transition | Shifiting | Correlation | N-Gau |
|---|---|---|---|---|---|---|---|---|---|---|---|---|
| ETTm1 | 7 | 57,600 | 6:2:2 | Electricity | 15 mins | 0.543 | 0.547 | 1.000 | 0.703 | 0.002 | 0.351 | 0.328 |
| ETTm2 | 7 | 57,600 | 6:2:2 | Electricity | 15 mins | 0.184 | 1.000 | 0.992 | 0.542 | 0.395 | 0.000 | 0.460 |
| ETTh1 | 7 | 14,400 | 6:2:2 | Electricity | 1 hour | 0.471 | 0.919 | 0.997 | 0.807 | 0.000 | 0.408 | 0.232 |
| ETTh2 | 7 | 14,400 | 6:2:2 | Electricity | 1 hour | 0.184 | 1.000 | 0.939 | 0.477 | 0.393 | 0.017 | 0.381 |
| Traffic | 862 | 17,544 | 7:1:2 | Traffic | 1 hour | 0.840 | 0.000 | 1.000 | 0.941 | 0.006 | 1.000 | 0.499 |
| Weather | 21 | 52,696 | 7:1:2 | Environment | 10 mins | 0.276 | 0.601 | 1.000 | 0.555 | 0.175 | 0.615 | 0.330 |
| Solar | 137 | 52,560 | 6:2:2 | Energy | 10 mins | 0.937 | 0.334 | 1.000 | 0.727 | 0.157 | 0.908 | 1.000 |
| Electricity | 321 | 26,304 | 7:1:2 | Electricity | 1 hour | 1.000 | 0.827 | 0.986 | 0.947 | 0.016 | 0.964 | 0.284 |
| Exchange | 8 | 7,588 | 7:1:2 | Economic | 1 day | 0.000 | 0.959 | 0.000 | 0.175 | 0.303 | 0.200 | 0.159 |
| ZafNoo | 11 | 19,225 | 7:1:2 | Nature | 30 mins | 0.537 | 0.633 | 0.879 | 0.590 | 0.019 | 0.306 | 0.499 |
| ILI | 7 | 966 | 7:1:2 | Health | 1 week | 0.665 | 0.702 | 0.530 | 0.540 | 0.758 | 0.551 | 0.103 |
| NASDAQ | 5 | 1,244 | 7:1:2 | Stock | 1 day | 0.688 | 0.976 | 0.530 | 0.000 | 1.000 | 0.194 | 0.100 |
| NN5 | 111 | 791 | 7:1:2 | Banking | 1 day | 0.595 | 0.103 | 0.919 | 1.000 | 0.154 | 0.677 | 0.000 |
| Wike2000 | 2,000 | 792 | 7:1:2 | Web | 1 day | 0.190 | 0.371 | 0.783 | 0.550 | 0.049 | 0.694 | 0.209 |

## A.2 TIME SERIES FORECASTING MODELS

In the realm of time series forecasting, numerous models have surfaced in recent years. We choose models with superior predictive performance in our benchmark, including the pre-trained time series models: ROSE (Wang et al., 2024c), TimesFM (Das et al., 2023), Timer (Liu et al., 2024b), TTM (Ekambaram et al., 2024), Moirai (Woo et al., 2024), and UniTS (Gao et al., 2024); The LLM-based models: GPT4TS (Zhou et al., 2024), S$^2$IPLLM (Pan et al., 2024), UniTime (Liu et al., 2024a) and Time-LLM (Jin et al., 2023a); And the specific models: TimesNet (Wu et al., 2022), Fedformer (Zhou et al., 2022), iTransformer (Liu et al., 2023), PatchTST (Nie et al., 2022), FITS (Xu et al., 2024), TimeMixer (Wang et al., 2024b), and Dlinear (Zeng et al., 2023). The specific descriptions for each of these models—see Table 9.

Table 9: Descriptions of time series forecasting models in `FoundTS`.

| Models | Descriptions |
|---|---|
| TimesFM | TimesFM is a decoder-only attention model for time-series forecasting, using input patching and trained on diverse real and synthetic data. It excels in zero-shot tasks across various datasets, forecast horizons, and time granularities. |
| Timer | Timer is a GPT-style autoregressive model for time series analysis, predicting the next token in single-series sequences. It supports tasks like forecasting, imputation, and anomaly detection across different time series. |
| UniTS | UniTS is a transformer-based model with task tokenization and dynamic self-attention across time and variables. It handles generative and predictive tasks across domains without needing task-specific modifications. |
| TTM | It is based on MLP-Mixer blocks with gated attention and multi-resolution sampling. It captures temporal patterns and cross-channel correlations for time-series forecasting, optimized for zero/few-shot learning with low computational cost. |
| Moment | Moment is a transformer system pre-trained on a masked time series task. It reconstructs masked portions of time series for tasks like forecasting, classification, anomaly detection, and imputation. |
| Moirai | Moirai is a masked encoder-based transformer using multi-patch projections and flexible attention to handle time series forecasting across various domains and frequencies. |
| ROSE | ROSE uses an encoder-decoder transformer with Decomposed Frequency Learning and a Time Series Register to separate temporal patterns and adaptively transfer across time series forecasting tasks. |
| GPT4TS | GPT4TS fine-tunes the limited parameters of LLM, which demonstrates competitive performance by transferring knowledge from large-scale pre-training text data. |
| S$^2$IPLLM | S$^2$IP-LLM aligns pre-trained language models with time series embeddings through tokenization and semantic anchors. It enhances forecasting by using semantic-informed prompting and cosine similarity. |
| UniTime | UniTime designs domain instructions to align time series and text modalit. |
| Time-LLM | Time-LLM reprograms time series into text to align the corresponding representation of LLMs to further activate the potential of LLMs. |
| PatchTST | PatchTST learns patch-wise dependencies, capturing more complex temporal dynamics and significantly improving forecasting performance. |
| DLinear | It employs a simple architecture with relatively few parameters and have also demonstrated good forecasting accuracy. |
| FITS | It operates on the principle that time series can be manipulated through interpolation in the complex frequency domain. |
| iTransformer | iTransformer applies attention and feed-forward networks to inverted dimensions, effectively considering the correlations among channels. |
| FEDFormer | FEDformer represents time series by randomly selecting a fixed number of Fourier components, covering both high- and low-frequency components. |
| TimesNet | TimesNet adaptively discovers multi-periodicity and captures complex temporal variations from transformed 2D tensors using a parameter-efficient inception block. |
| TimeMixer | It employs a fully MLP-based architecture, utilizing Past-Decomposable-Mixing and Future-Multipredictor-Mixing blocks to fully leverage disentangled multiscale time series during both the past extraction and future prediction phases. |

## A.3 IMPLEMENTATION DETAILS

All experiments are conducted using PyTorch (Paszke et al., 2019) in Python 3.10 and execute on an NVIDIA Tesla-A800 GPU. The training process is guided by the L2 loss, employing the ADAM (Kingma, 2014) optimizer. Initially, the batch size is set to 64, with the option to reduce it by half (to a minimum of 8) in case of an Out-Of-Memory (OOM) situation. The initial learning rate is set to 0.0001 and dynamically adjusted through a simulated annealing approach over a total of 20 training epochs. Additionally, to mitigate the risk of overfitting, we implemented an early stopping strategy with a patience parameter set to 3.

## B TIME SERIES CHARACTERISTICS

### B.1 TREND

The trend of a time series refers to the long-term changes or patterns that occur over time. Intuitively, it represents the general direction in which the data is moving. Referring to the explained variance O'Grady (1982), Trend Strength can be defined as in Algorithm 1. Seasonal and Trend decomposition using Loess (STL), which is a highly versatile and robust method for time series decomposition Cleveland et al. (1990)

---

**Algorithm 1** Calculating Trend Values of Time Series

---

**Input:** Time series $X \in \mathbb{R}^{T \times 1}$
**Output:** $Trend\_Strength$ $\beta \in (0, 1)$ of $X$

1: $S, T, R \leftarrow STL(X); X = S + T + R$
2: **return** $\beta \leftarrow max\left(0, 1 - \frac{var(R)}{var(T+R)}\right)$

---

### B.2 SEASONALITY

Seasonality refers to the phenomenon where changes in a time series repeat at specific intervals. Algorithm 2 details the calculation process.

---

**Algorithm 2** Calculating Seasonality Values of Time Series

---

**Input:** Time series $X \in \mathbb{R}^{T \times 1}$
**Output:** $Seasonality\_Strength$ $\zeta \in (0, 1)$ of $X$

1: $S, T, R \leftarrow STL(X); X = S + T + R$
2: **return** $\zeta \leftarrow \max\left(0, 1 - \frac{var(R)}{var(S+R)}\right)$

---

### B.3 STATIONARITY

Stationarity refers to the mean of any observation in a time series $X = \langle x_1, x_2, ..., x_n \rangle$ is constant, and the variance is finite for all observations. Also, the covariance $cov(x_i, x_j)$ between any two observations $x_i$ and $x_j$ depends only on their distance $|j - i|$, i.e., $\forall i + r \leq n, j + r \leq n$ $(cov(x_i, x_j) = cov(x_{i+r}, x_{j+r}))$. Strictly stationary time series are rare in practice. Therefore, weak stationarity conditions are commonly applied Lee (2017) Nason (2006). In our paper, we also exclusively focus on weak stationarity.

We adopt the Augmented Dick-Fuller (ADF) test statistic Elliott et al. (1992) to quntify stationarity. Algorithm 3 details the calculation process.

---

**Algorithm 3** Calculating Stationarity Values of Time Series

---

**Input:** Time series $X \in \mathbb{R}^{T \times 1}$
**Output:** Stationarity value $\gamma \in \{0, 1\}$ of $X$

1: $s \leftarrow ADF(X)$
2: **return** $\gamma \leftarrow (s <= 0.05)$

---

### B.4 SHIFTING

Shifting refers to the phenomenon where the probability distribution of time series changes over time. This behavior can stem from structural changes within the system, external influences, or the occurrence of random events. As the value approaches 1, the degree of shifting becomes more severe. Algorithm 4 details the calculation process.

---

**Algorithm 4** Calculating Shifting Values of Time Series

---

**Input:** Time series $X \in \mathbb{R}^{T \times 1}$
**Output:** Shifting value $\delta \in (0, 1)$ of $X$

1: Normalize $X$ by calculating the z-score to obtain $Z \in \mathbb{R}^{T \times 1}$
2: $Z_{\min} \leftarrow \min(Z)$, $Z_{\max} \leftarrow \max(Z)$
3: $S \leftarrow \{s_i \mid s_i \leftarrow Z_{\min} + (i-1)\frac{Z_{\max} - Z_{\min}}{m}, 1 \leq i \leq m\}$ where m is the number of thresholds
4: **for** $s_i$ in $S$ **do**
5:    $K \leftarrow \{j \mid Z_j > s_i, 1 \leq j \leq T\}$, $M_i \leftarrow median(K)$, $1 \leq i \leq m$
6: **end for**
7: $M' \leftarrow Min-Max\ Normalization(M)$
8: **return** $\delta \leftarrow median(\{M'_1, M'_2, ..., M'_m\})$

---

### B.5 TRANSITION

Transition refers to the trace of the covariance of transition matrix between symbols in a 3-letter alphabet Lubba et al. (2019). It captures the regular and identifiable fixed features present in a time series, such as the clear manifestation of trends, periodicity, or the simultaneous presence of both seasonality and trend. Algorithm 5 details the calculation process.

---

**Algorithm 5** Calculating Transition Values of Time Series

---

**Input:** Time series $X \in \mathbb{R}^{T \times 1}$
**Output:** Transition value $\Delta \in (0, \frac{1}{3})$ of $X$

1: Calculate the first zero crossing of the autocorrelation function:
   $\tau \leftarrow firstzero\_ac(X)$
2: Generate $Y \in \mathbb{R}^{T' \times 1}$ by downsampling X with stride $\tau$
3: Define index $I = argsort(Y) \in \mathbb{R}^{T' \times 1}$, then characterize $Y$ to obtain $Z \in \mathbb{R}^{T' \times 1}$:
4: **for** $j \in [0 : T']$ **do**
5:    $Z[j] \leftarrow floor(\ I[j]/\frac{1}{3}T')$
6: **end for**
7: Generate a transition matrix $M \in \mathbb{R}^{3 \times 3}$:
8: **for** $j \in [0 : T']$ **do**
9:    $M[Z[j] - 1][Z[j+1] - 1]$++
10: **end for**
11: $M' \leftarrow \frac{1}{T'} M$
12: Compute the covariance matrix $C$ between the columns of $M'$
13: **return** $\Delta \leftarrow tr(C)$

---

## B.6 CORRELATION

Correlation refers to the possibility that different variables in a multivariate time series may share common trends or patterns, indicating that they are influenced by similar factors or have some underlying relationship. Algorithm 6 details the calculation process. Catch22 Lubba et al. (2019) is a library designed to extract 22 distinct features from time series data, facilitating comprehensive analysis and understanding of temporal patterns.

---

**Algorithm 6** Calculating Correlation Values of Time Series

---

**Input:** Time series $X \in \mathbb{R}^{T \times N}$
**Output:** Correlation value $\Delta \in (0, 1)$ of $X$

1: Get the representation for each channel using the Catch22 library:
    $F = \langle F^1, F^2, ..., F^N \rangle \in \mathbb{R}^{22 \times N} \leftarrow Catch22(X)$
2: Calculate the Pearson correlation coefficients between all pairs of channels:
    $P = \left\{ r(F^i, F^j) \mid 1 \leq i \leq N, i + 1 \leq j \leq N, i, j \in N^* \right\}$
3: Compute the correlation by computing the mean and variance of all Pearson correlation coefficients (PCCs)
    $Correlation = mean\,(P) + \frac{1}{1 + var(P)}$
4: **return** $Correlation$

---

## B.7 NON-GAUSSIANITY

Non-Gaussianity complexity refers to the extent to which the distribution of values within a time series segment deviates from a Gaussian distribution, measuring the intricacy and variability of the data distribution. Algorithm 7 details the calculation process.

---

**Algorithm 7** Calculating Non-Gaussianity of Time Series

---

**Input:** Time series $X \in \mathbb{R}^{T \times 1}$, window length $w$
**Output:** Average non-Gaussianity $avg\_JSD$ of $X$

1: **function** $JSD(P, Q)$
2:     $M \leftarrow 0.5 \times (P + Q)$
3:     $kl\_p\_m \leftarrow KL\_divergence(P, M)$
4:     $kl\_q\_m \leftarrow KL\_divergence(Q, M)$
5:     **return** $0.5 \times (kl\_p\_m + kl\_q\_m)$
6: **end function**

7: Divide $X$ into windows $P_1, P_2, \ldots, P_n$ where $P_i \in \mathbb{R}^{w \times 1}$
8: Initialize total_JSD $\leftarrow 0$
9: **for** each window $P_i$ **do**
10:     Fit a Gaussian distribution $Q_i$ to $P_i$
11:     Calculate the JS Divergence JSD$(P_i, Q_i)$
12:     total_JSD $\leftarrow$ total_JSD $+$ JSD$(P_i, Q_i)$
13: **end for**
14: avg_JSD $\leftarrow \frac{\text{total\_JSD}}{n}$
15: **return** avg_JSD

---

## C  MORE ANALYSIS ON DIFFERENT FOUNDATION MODELS

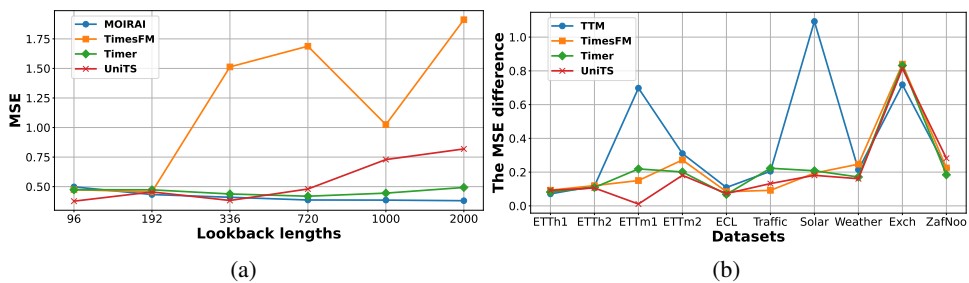

|           |           |
|:---------:|:---------:|
| (a)       | (b)       |

Figure 6: (a) Reports the MSE zero-shot results for predicting 96 length on the ETTh1 dataset with different lookback lengths. (b) Reports the zero-shot MSE difference between predicting 720 length and 96 length across different datasets.

### C.1  EFFECTIVENESS OF LOOKBACK LENGTHS AND PREDICTION LENGTHS

To investigate whether the length of the lookback (i.e., the amount of historical information received by the model) affects its performance and whether the model can flexibly predict different lengths, we conducted an analysis experiment. Figure 6a illustrates that MOIRAI's performance steadily improves as the look-back length increases. In contrast, the performance of other models does not consistently enhance with longer look-back lengths and may occasionally decline significantly. This suggests that when designing models, we should ensure that they can flexibly handle varying lookback lengths and effectively utilize more historical information.

Next, we study the effects on prediction lengths. We report the MSE differences when predicting lengths are 720 vs. 96 in Figure 6b. The results show that models like TTM, which cannot handle arbitrary prediction lengths, exhibit larger fluctuations in MSE differences than other models that can output predictions at arbitrary lengths. This highlights the need to design models that can flexibly predict across different prediction lengths.

### C.2  PRE-TRAINED DATA ANALYSIS

To analyze whether the domain of pre-trained time series data affects downstream predictions, we summarize the pre-training datasets used by various models, as shown in Table 10. Combining the results from zero-shot and few-shot forecasting in Tables 4 and 5, we draw the following conclusions: 1) ROSE utilizes a broader range of pre-training data domains, achieving good or best prediction results on some datasets with small-sized model parameter and pre-training datasets. This suggests that the domain diversity of pre-training data may be crucial for pre-trained models. 2) Some pre-trained models achieve good results when the downstream domain is not included in the pre-trained data, indicating that pre-trained models may possess generalization potential for unseen data domains. For instance, TimesFM and MOIRAI perform well on the ILI and Exchange datasets in a zero-shot setting, despite the absence of Health and Finance domains in pre-training data. 3) The presence of downstream domains in the pre-training data does not guarantee good prediction performance. For example, UniTS does not exhibit good performance on the Solar dataset, though the model uses energy domain data in pretraining. This indicates that the model's prediction performance depends not only on the diversity of the pre-training data but also on other factors.

Table 10: The domain of the pre-trained data for different foundation models.

| Domains     | TimesFM | Timer | MOIRAI | TTM | UniTS | Moment | ROSE |
|-------------|:-------:|:-----:|:------:|:---:|:-----:|:------:|:----:|
| Nature      | ✓       | ✓     | ✓      |     |       | ✓      | ✓    |
| Traffic     | ✓       | ✓     |        | ✓   | ✓     | ✓      | ✓    |
| Energy      | ✓       |       | ✓      | ✓   | ✓     | ✓      | ✓    |
| Web         | ✓       | ✓     |        | ✓   |       | ✓      | ✓    |
| Health      |         | ✓     |        | ✓   | ✓     | ✓      | ✓    |
| Environment | ✓       | ✓     | ✓      |     | ✓     | ✓      | ✓    |
| Electricity | ✓       |       |        | ✓   | ✓     | ✓      | ✓    |
| Banking     |         |       |        | ✓   | ✓     |        | ✓    |
| Stock       |         |       |        |     | ✓     |        | ✓    |
| Economic    |         | ✓     |        | ✓   | ✓     |        | ✓    |

# D   FULL RESULTS

To comprehensively and fairly compare foundation models with time series specific models, we conduct experiments across datasets from different domains using zero-shot, few-shot, and full-shot settings. The zero-shot experiments assess the generalization ability of foundation models to new data, with results presented in Tables 11 and 12. The few-shot experiments use 5% of the training data for fine-tuning, evaluating whether foundation models can generalize effectively with limited data, as shown in Tables 13 and 14. The full-shot experiments verify the optimal performance of foundation models under full data conditions, with results in Tables 15 and 16. For lookback lengths, we select 36 and 104 for NASDAQ, NN5, ILI, and Wiki2000 datasets, and 96, 336, and 512 for all other datasets. For prediction lengths, we choose 24, 36, 48, and 60 for NASDAQ, NN5, ILI, and Wiki2000, and 96, 192, 336, and 720 for the rest. The reported results reflect the best performances across different lookback lengths.

## D.1   ZERO-SHOT RESULTS

Table 11: Pre-trained model results in the **zero-shot setting**. The results are **MSE** of each prediction length.

| Model | Horizon | ETTh1 | ETTh2 | ETTm1 | ETTm2 | Electricity | Traffic | Solar | Weather | Exchange | ZafNoo | ILI | NASDAQ | NN5 | Wike2000 |
|---|---|---|---|---|---|---|---|---|---|---|---|---|---|---|---|
| TimesFM | 96 | 0.421 | 0.326 | **0.363** | 0.206 | **0.119** | **0.327** | 0.408 | **0.123** | 0.096 | 0.517 | **2.580** | 0.664 | **0.830** | **414.160** |
| | 192 | 0.472 | 0.397 | **0.417** | 0.293 | **0.137** | **0.354** | 0.466 | **0.170** | 0.195 | 0.604 | 2.979 | 1.032 | **0.774** | **471.320** |
| | 336 | 0.510 | 0.431 | **0.447** | 0.411 | **0.157** | **0.378** | 0.526 | **0.240** | 0.332 | 0.660 | 3.328 | 1.193 | 0.755 | 498.943 |
| | 720 | 0.514 | 0.446 | **0.513** | 0.478 | **0.203** | **0.420** | 0.601 | **0.370** | 0.935 | 0.742 | **3.212** | 1.247 | 0.760 | 517.905 |
| Timer | 96 | 0.414 | 0.305 | 0.440 | **0.203** | 0.221 | 0.526 | 0.549 | 0.178 | **0.095** | 0.467 | 2.632 | 0.609 | 1.184 | 528.471 |
| | 192 | 0.440 | 0.365 | 0.505 | **0.265** | 0.246 | 0.561 | 0.631 | 0.228 | 0.198 | 0.531 | 2.645 | 0.886 | 1.139 | 585.102 |
| | 336 | 0.455 | 0.378 | 0.570 | 0.319 | 0.272 | 0.614 | 0.702 | 0.281 | 0.349 | 0.579 | 2.668 | 0.976 | 1.126 | 625.590 |
| | 720 | 0.496 | 0.414 | 0.659 | 0.405 | 0.288 | 0.749 | 0.757 | 0.349 | 0.927 | 0.651 | 6.147 | 1.087 | 1.589 | 681.115 |
| UniTS | 96 | 0.377 | 0.323 | 0.761 | 0.249 | 0.175 | 0.481 | 0.771 | 0.194 | 0.130 | 0.570 | 4.407 | 1.149 | 1.303 | 528.415 |
| | 192 | 0.398 | 0.372 | 0.777 | 0.309 | 0.178 | 0.447 | 0.800 | 0.252 | 0.232 | 0.610 | 4.396 | 1.265 | 1.279 | 577.330 |
| | 336 | 0.413 | 0.373 | 0.754 | 0.353 | 0.190 | 0.445 | 0.855 | 0.299 | 0.386 | 0.833 | 4.336 | 1.232 | 1.291 | 678.552 |
| | 720 | 0.469 | 0.429 | 0.750 | 0.430 | 0.248 | 0.613 | 0.952 | 0.355 | 0.943 | 0.852 | 4.316 | 1.133 | 1.294 | 667.596 |
| TTM | 96 | **0.363** | 0.286 | 0.415 | **0.186** | 0.170 | 0.509 | **0.193** | 0.152 | **0.084** | 0.427 | 4.750 | 1.214 | 1.371 | 442.993 |
| | 192 | **0.392** | 0.343 | 0.476 | 0.265 | 0.183 | 0.524 | **0.216** | 0.173 | 0.197 | 0.494 | 4.913 | 1.554 | 1.330 | 500.224 |
| | 336 | 0.423 | 0.365 | 1.113 | 0.407 | 0.244 | 0.696 | 1.404 | 0.294 | 0.311 | 0.571 | 4.375 | 1.869 | 1.317 | 522.895 |
| | 720 | 0.434 | 0.403 | 1.113 | 0.496 | 0.279 | 0.714 | 1.286 | 0.367 | **0.802** | 0.649 | 4.343 | 2.032 | 1.313 | 545.447 |
| MOIRAI | 96 | 0.394 | **0.285** | 0.516 | 0.222 | 0.212 | 1.359 | 0.767 | 0.208 | 0.096 | 0.441 | 2.929 | **0.563** | 0.855 | - |
| | 192 | 0.430 | 0.352 | 0.536 | 0.303 | 0.225 | 1.387 | 0.777 | 0.281 | 0.197 | 0.499 | 3.385 | **0.957** | 0.786 | - |
| | 336 | 0.450 | 0.384 | 0.564 | 0.366 | 0.245 | - | 0.790 | **0.340** | 0.349 | 0.543 | 3.639 | 1.294 | 0.758 | - |
| | 720 | 0.449 | 0.418 | 0.631 | 0.456 | 0.282 | - | 0.808 | 0.420 | 0.903 | 0.606 | 3.676 | 1.366 | **0.745** | - |
| ROSE | 96 | **0.382** | 0.298 | 0.512 | 0.224 | 0.209 | 0.572 | 0.537 | 0.200 | 0.266 | 0.481 | 4.790 | 0.807 | 1.368 | 578.651 |
| | 192 | 0.400 | **0.336** | 0.512 | 0.266 | 0.219 | 0.575 | 0.517 | 0.239 | 0.393 | 0.527 | 4.780 | 1.140 | 1.320 | 637.540 |
| | 336 | **0.404** | **0.336** | 0.523 | 0.310 | 0.236 | 0.588 | **0.517** | 0.279 | 0.587 | 0.562 | 4.570 | 1.293 | 1.313 | 669.432 |
| | 720 | **0.420** | **0.395** | 0.552 | **0.395** | 0.273 | 0.618 | **0.517** | 0.340 | 1.227 | **0.545** | 4.270 | 1.282 | 1.317 | 712.387 |

MOIRAI flattens all channels into a single dimension for patching, thus limiting its use when dealing with datasets with many channels. MOIRAI fails to work on Traffic (862 channels) and Wike2000 (2000 channels), which is shown in –.

Table 12: Pre-trained model results in the **zero-shot setting**. The results are **MAE** of each prediction length.

| Model | Horizon | ETTh1 | ETTh2 | ETTm1 | ETTm2 | Electricity | Traffic | Solar | Weather | Exchange | ZafNoo | ILI | NASDAQ | NN5 | Wike2000 |
|---|---|---|---|---|---|---|---|---|---|---|---|---|---|---|---|
| TimesFM | 96 | 0.401 | 0.351 | **0.369** | **0.267** | **0.212** | **0.220** | 0.345 | **0.159** | 0.215 | 0.412 | **1.008** | 0.546 | **0.609** | **0.945** |
| | 192 | 0.432 | 0.396 | **0.405** | **0.320** | **0.229** | **0.235** | 0.373 | **0.204** | 0.313 | 0.464 | 1.128 | 0.695 | **0.600** | **1.081** |
| | 336 | 0.455 | 0.428 | **0.428** | 0.414 | **0.248** | **0.248** | 0.407 | **0.261** | 0.416 | 0.496 | 1.208 | 0.754 | **0.602** | **1.137** |
| | 720 | 0.481 | 0.454 | **0.470** | 0.437 | **0.287** | **0.272** | 0.461 | **0.352** | 0.723 | 0.542 | **1.189** | 0.778 | **0.609** | **1.145** |
| Timer | 96 | 0.439 | 0.355 | 0.422 | 0.285 | 0.322 | 0.368 | 0.487 | 0.227 | 0.219 | 0.418 | 1.082 | 0.559 | 0.835 | 1.243 |
| | 192 | 0.455 | 0.400 | 0.458 | 0.327 | 0.342 | 0.385 | 0.547 | 0.274 | 0.322 | 0.456 | 1.098 | 0.685 | 0.824 | 1.362 |
| | 336 | 0.463 | 0.413 | 0.490 | 0.361 | 0.361 | 0.410 | 0.596 | 0.313 | 0.431 | 0.482 | 1.105 | 0.724 | 0.822 | 1.461 |
| | 720 | 0.496 | 0.444 | 0.534 | 0.410 | 0.374 | 0.464 | 0.646 | 0.364 | 0.729 | 0.519 | 1.861 | 0.783 | 1.030 | 1.644 |
| UniTS | 96 | **0.392** | 0.355 | 0.530 | 0.315 | 0.269 | 0.328 | 0.594 | 0.234 | 0.255 | 0.531 | 1.553 | 0.768 | 0.913 | 1.161 |
| | 192 | 0.421 | 0.406 | 0.534 | 0.352 | 0.273 | 0.307 | 0.618 | 0.279 | 0.346 | 0.551 | 1.548 | 0.864 | 0.912 | 1.237 |
| | 336 | **0.425** | 0.413 | 0.539 | 0.383 | 0.287 | 0.299 | 0.672 | 0.316 | 0.452 | 0.687 | 1.500 | 0.848 | 0.915 | 1.601 |
| | 720 | 0.463 | 0.457 | 0.569 | 0.431 | 0.335 | 0.381 | 0.793 | 0.361 | 0.738 | 0.704 | 1.503 | 0.817 | 0.919 | 1.619 |
| TTM | 96 | 0.396 | 0.343 | 0.416 | 0.271 | 0.265 | 0.343 | **0.256** | 0.199 | **0.203** | 0.387 | 1.550 | 0.868 | 0.939 | 1.347 |
| | 192 | **0.415** | 0.384 | 0.456 | 0.322 | 0.278 | 0.351 | **0.271** | 0.242 | 0.297 | 0.429 | 1.572 | 0.961 | 0.925 | 1.426 |
| | 336 | 0.430 | 0.412 | 0.711 | 0.426 | 0.322 | 0.424 | 0.876 | 0.333 | 0.406 | 0.472 | 1.452 | 1.021 | 0.918 | 1.457 |
| | 720 | 0.451 | 0.439 | 0.725 | 0.470 | 0.353 | 0.432 | 0.820 | 0.377 | **0.668** | 0.511 | 1.448 | 1.042 | 0.916 | 1.498 |
| MOIRAI | 96 | 0.399 | **0.329** | 0.431 | 0.282 | 0.301 | 0.789 | 0.716 | 0.221 | 0.213 | 0.391 | 1.113 | **0.518** | 0.644 | - |
| | 192 | 0.422 | **0.373** | 0.446 | 0.348 | 0.320 | 0.798 | 0.722 | 0.270 | 0.312 | **0.429** | 1.207 | **0.673** | 0.623 | - |
| | 336 | 0.437 | 0.402 | 0.460 | 0.373 | 0.333 | - | 0.730 | 0.313 | 0.425 | 0.451 | 1.257 | 0.792 | 0.615 | - |
| | 720 | 0.450 | **0.431** | 0.490 | 0.428 | 0.358 | - | 0.738 | 0.370 | 0.717 | 0.478 | 1.254 | 0.819 | 0.611 | - |
| ROSE | 96 | 0.408 | 0.362 | 0.460 | 0.309 | 0.307 | 0.407 | 0.564 | 0.260 | 0.385 | 0.445 | 1.630 | 0.684 | 0.952 | 1.270 |
| | 192 | 0.420 | 0.385 | 0.462 | 0.333 | 0.315 | 0.406 | 0.556 | 0.288 | 0.472 | 0.470 | 1.630 | 0.803 | 0.937 | 1.360 |
| | 336 | 0.426 | **0.399** | 0.470 | 0.358 | 0.330 | 0.411 | **0.559** | 0.315 | 0.587 | 0.488 | 1.580 | 0.842 | 0.936 | 1.430 |
| | 720 | **0.447** | 0.432 | 0.490 | **0.407** | 0.328 | 0.422 | **0.540** | 0.357 | 0.832 | 0.512 | 1.520 | 0.835 | 0.940 | 1.520 |

MOIRAI flattens all channels into a single dimension for patching, thus limiting its use when dealing with datasets with many channels. MOIRAI fails to work on Traffic (862 channels) and Wike2000 (2000 channels), which is shown with –.

## D.2 FEW-SHOT RESULTS

Table 13: Model results in the **5% few-shot setting**. The results are **MSE** of each prediction length.

| | Model | Horizon | ETTh1 | ETTh2 | ETTm1 | ETTm2 | Electricity | Traffic | Solar | Weather | Exchange | ZafNoo |
|---|---|---|---|---|---|---|---|---|---|---|---|---|
| TS Pretrain Model | TimesFM | 96 | 0.435 | 0.315 | 0.412 | 0.242 | 0.229 | 0.638 | 0.340 | 0.233 | 0.119 | 0.574 |
| | | 192 | 0.453 | 0.356 | 0.477 | 0.303 | 0.253 | 0.614 | 0.510 | 0.336 | 0.210 | 0.651 |
| | | 336 | 0.494 | 0.362 | 1.156 | 0.380 | 0.296 | 0.595 | 0.633 | 0.393 | 0.672 | 0.756 |
| | | 720 | 0.453 | 0.396 | 1.096 | 0.476 | 0.363 | 1.025 | 0.754 | 0.411 | 0.881 | 0.828 |
| | Timer | 96 | 0.371 | 0.283 | **0.288** | 0.168 | 0.142 | **0.389** | **0.174** | **0.150** | 0.083 | 0.430 |
| | | 192 | 0.399 | 0.349 | 0.330 | 0.235 | 0.159 | 0.408 | **0.193** | **0.196** | 0.173 | **0.487** |
| | | 336 | 0.413 | 0.366 | **0.363** | 0.289 | 0.178 | 0.423 | 0.209 | 0.249 | 0.312 | **0.527** |
| | | 720 | 0.443 | 0.400 | 0.422 | 0.379 | 0.221 | 0.459 | 0.231 | 0.328 | 0.826 | 0.591 |
| | UniTS | 96 | 0.378 | 0.314 | 0.453 | 0.200 | **0.140** | 0.398 | **0.204** | 0.161 | 0.116 | 0.476 |
| | | 192 | 0.397 | 0.361 | 0.517 | 0.269 | 0.157 | 0.410 | 0.235 | 0.213 | 0.226 | 0.540 |
| | | 336 | 0.437 | 0.362 | 0.600 | 0.326 | 0.172 | 0.424 | 0.249 | 0.267 | 0.376 | 0.590 |
| | | 720 | 0.494 | 0.437 | 0.635 | 0.412 | 0.210 | 0.471 | 0.239 | 0.336 | 0.932 | 0.742 |
| | TTM | 96 | **0.361** | 0.283 | 0.330 | 0.166 | 0.154 | 0.454 | 0.189 | 0.153 | **0.082** | **0.424** |
| | | 192 | **0.390** | 0.338 | 0.367 | 0.225 | 0.169 | 0.471 | 0.195 | 0.198 | 0.170 | 0.487 |
| | | 336 | 0.420 | 0.361 | 1.197 | 0.367 | 0.262 | 0.790 | 1.624 | 0.302 | **0.303** | 0.558 |
| | | 720 | 0.429 | 0.399 | 1.152 | 0.453 | 0.295 | 0.803 | 1.522 | 0.369 | **0.785** | 0.624 |
| | Moment | 96 | 0.441 | 0.328 | 0.331 | 0.192 | 0.198 | 0.795 | 0.521 | 0.182 | 0.129 | 0.516 |
| | | 192 | 0.455 | 0.362 | 0.356 | 0.238 | 0.229 | 0.753 | 0.538 | 0.222 | 0.226 | 0.502 |
| | | 336 | 0.459 | 0.368 | 0.380 | 0.285 | 0.279 | 0.759 | 0.544 | 0.264 | 0.375 | 0.567 |
| | | 720 | 0.517 | 0.418 | 0.428 | 0.367 | 0.278 | 0.796 | 0.515 | 0.328 | 0.930 | 0.597 |
| | MOIRAI | 96 | 0.401 | 0.300 | 0.451 | 0.241 | 0.205 | 0.507 | 8.002 | 0.215 | 0.097 | 0.673 |
| | | 192 | 0.454 | 0.385 | 0.474 | 0.321 | 0.216 | 0.540 | 7.045 | 0.263 | 0.202 | 0.890 |
| | | 336 | 0.473 | 0.393 | 0.512 | 0.409 | 0.228 | - | 4.575 | 0.312 | 0.386 | 0.944 |
| | | 720 | 0.479 | 0.509 | 0.648 | 0.571 | 0.263 | - | 2.494 | 0.382 | 0.915 | 1.178 |
| | ROSE | 96 | 0.371 | **0.272** | **0.291** | **0.163** | 0.143 | 0.392 | 0.192 | 0.159 | 0.118 | 0.453 |
| | | 192 | 0.398 | 0.334 | **0.291** | **0.217** | 0.163 | 0.409 | 0.204 | 0.206 | 0.324 | 0.520 |
| | | 336 | **0.406** | 0.360 | 0.370 | **0.357** | 0.186 | **0.415** | 0.204 | 0.263 | 0.440 | 0.562 |
| LLM | GPT4TS | 96 | 0.438 | 0.322 | 0.342 | 0.189 | 0.177 | 0.402 | 0.226 | 0.187 | 0.116 | 0.515 |
| | | 192 | 0.454 | 0.363 | 0.369 | 0.237 | 0.191 | 0.413 | 0.253 | 0.225 | 0.219 | 0.552 |
| | | 336 | 0.461 | 0.378 | 0.393 | 0.290 | 0.209 | 0.422 | 0.262 | 0.268 | 0.375 | 0.582 |
| | | 720 | 0.509 | 0.427 | 0.440 | 0.374 | 0.246 | 0.459 | 0.265 | 0.329 | 0.944 | 0.610 |
| | S2IPLLM | 96 | 0.665 | 0.366 | 0.361 | 0.215 | - | - | 0.229 | 0.171 | 0.110 | 0.655 |
| | | 192 | 0.676 | 0.381 | 0.390 | 0.276 | - | - | 0.305 | 0.219 | 0.210 | 0.684 |
| | | 336 | 0.672 | 0.392 | 0.419 | 0.320 | - | - | 0.361 | 0.261 | 0.358 | 0.746 |
| | | 720 | 0.717 | 0.444 | 0.462 | 0.401 | - | - | 0.361 | 0.330 | 0.951 | 0.948 |
| | TimeLLM | 96 | 0.539 | 0.342 | 0.314 | 0.183 | - | - | - | 0.170 | 0.116 | 0.561 |
| | | 192 | 0.702 | 0.402 | 0.352 | 0.235 | - | - | - | 0.205 | 0.182 | 0.530 |
| | | 336 | 0.716 | 0.388 | 0.389 | 0.284 | - | - | - | 0.253 | 0.382 | 0.645 |
| | | 720 | 0.739 | 0.451 | 0.438 | 0.376 | - | - | - | **0.322** | 0.950 | 0.663 |
| | UniTime | 96 | 0.716 | 0.358 | 0.356 | 0.188 | 0.174 | 0.436 | 0.216 | 0.184 | 0.113 | 0.679 |
| | | 192 | 0.801 | 0.419 | 0.393 | 0.242 | 0.189 | **0.399** | 0.209 | 0.271 | 0.363 | 0.790 |
| | | 336 | 0.820 | 0.430 | 0.420 | 0.291 | 0.426 | 0.426 | 0.228 | 0.328 | 0.379 | 0.853 |
| | | 720 | 0.860 | 0.476 | 0.468 | 0.373 | 0.437 | 0.473 | **0.220** | 0.395 | 0.978 | 0.891 |
| Specific Model | PatchTST | 96 | 0.404 | 0.345 | 0.314 | 0.195 | 0.158 | 0.409 | 0.254 | 0.172 | 0.095 | 0.499 |
| | | 192 | 0.431 | 0.377 | 0.348 | 0.245 | 0.163 | 0.422 | 0.269 | 0.218 | 0.187 | 0.528 |
| | | 336 | 0.440 | 0.384 | 0.397 | 0.293 | 0.180 | 0.430 | 0.277 | 0.265 | 0.318 | 0.559 |
| | | 720 | 0.572 | 0.437 | 0.436 | 0.385 | 0.216 | 0.467 | 0.274 | 0.325 | 0.917 | 0.591 |
| | DLinear | 96 | 0.391 | 0.377 | 0.341 | 0.186 | 0.142 | 0.416 | 0.217 | 0.185 | 0.096 | 0.446 |
| | | 192 | 0.427 | 0.448 | 0.366 | 0.285 | **0.155** | 0.429 | 0.244 | 0.225 | 0.216 | 0.495 |
| | | 336 | 0.452 | 0.518 | 0.385 | 0.343 | **0.171** | 0.443 | 0.263 | 0.266 | 0.337 | 0.528 |
| | | 720 | 0.565 | 0.655 | 0.433 | 0.372 | 0.205 | 0.473 | 0.265 | 0.323 | 0.796 | 0.557 |
| | FITS | 96 | 0.376 | 0.277 | 0.303 | 0.165 | 0.146 | 0.407 | 0.208 | 0.172 | **0.082** | 0.454 |
| | | 192 | 0.400 | **0.331** | 0.337 | 0.219 | 0.160 | 0.418 | 0.229 | 0.215 | 0.173 | 0.509 |
| | | 336 | 0.419 | 0.350 | 0.368 | 0.272 | 0.178 | 0.433 | 0.241 | 0.261 | 0.317 | 0.550 |
| | | 720 | 0.435 | **0.382** | 0.420 | 0.359 | 0.219 | 0.486 | 0.248 | 0.326 | 0.825 | 0.593 |
| | iTransformer | 96 | 0.515 | 0.334 | 0.328 | 0.197 | 0.231 | 0.437 | 0.228 | 0.190 | 0.126 | 0.601 |
| | | 192 | 0.534 | 0.393 | 0.369 | 0.254 | 0.262 | 0.459 | 0.260 | 0.234 | 0.228 | 0.673 |
| | | 336 | 0.670 | 0.428 | 0.449 | 0.306 | 0.184 | 0.482 | 0.249 | 0.294 | 0.375 | 0.730 |
| | | 720 | 0.571 | 0.488 | 0.513 | 0.392 | 0.230 | 0.537 | 0.269 | 0.357 | 0.936 | 0.792 |
| | FEDformer | 96 | 0.476 | 0.400 | 0.521 | 0.249 | 0.272 | 0.763 | 0.796 | 0.245 | 0.167 | 0.529 |
| | | 192 | 0.533 | 0.494 | 0.667 | 0.310 | 0.289 | 0.884 | 0.639 | 0.271 | 0.553 | 0.609 |
| | | 336 | 0.606 | 0.407 | 0.731 | 0.481 | 0.310 | 0.935 | 0.739 | 0.352 | 0.721 | 0.691 |
| | | 720 | 0.613 | 0.607 | 0.746 | 0.631 | 0.339 | 0.761 | 0.668 | 0.587 | 1.133 | 0.902 |
| | TimesNet | 96 | 0.840 | 0.380 | 0.408 | 0.198 | 0.247 | 0.769 | 0.342 | 0.200 | 0.140 | 0.606 |
| | | 192 | 0.817 | 0.461 | 0.436 | 0.304 | 0.250 | 0.852 | 0.369 | 0.245 | 0.248 | 0.551 |
| | | 336 | 0.879 | 0.448 | 0.521 | 0.325 | 0.260 | 0.858 | 0.414 | 0.301 | 0.391 | 0.632 |
| | | 720 | 0.882 | 0.508 | 0.567 | 0.465 | 0.294 | 1.005 | 0.385 | 0.391 | 0.973 | 0.658 |
| | TimeMixer | 96 | 0.449 | 0.300 | 0.311 | 0.182 | 0.169 | 0.423 | 0.208 | 0.161 | 0.091 | 0.447 |
| | | 192 | 0.743 | 0.348 | 0.358 | 0.246 | 0.182 | 0.488 | 0.223 | 0.209 | 0.179 | 0.510 |
| | | 336 | 0.730 | 0.378 | 0.384 | 0.290 | 0.216 | 0.467 | 0.240 | **0.247** | 0.315 | 0.558 |
| | | 720 | 0.585 | 0.433 | 0.451 | 0.375 | 0.241 | 0.522 | 0.228 | 0.325 | 0.887 | 0.615 |

The maximum training duration was constrained to a maximum of 5 hours. Models that exceeded this threshold are represented with –.

Table 14: Model results in the **5% few-shot setting**. The results are **MAE** of each prediction length.

| | Model | Horizon | ETTh1 | ETTh2 | ETTm1 | ETTm2 | Electricity | Traffic | Solar | Weather | Exchange | ZafNoo |
|---|---|---|---|---|---|---|---|---|---|---|---|---|
| TS Pretrain Model | TimesFM | 96 | 0.416 | 0.366 | 0.409 | 0.309 | 0.311 | 0.340 | 0.359 | 0.296 | 0.248 | 0.493 |
| | | 192 | 0.431 | 0.392 | 0.444 | 0.348 | 0.332 | 0.326 | 0.457 | 0.336 | 0.331 | 0.526 |
| | | 336 | 0.449 | 0.403 | 0.689 | 0.401 | 0.369 | 0.316 | 0.521 | 0.402 | 0.632 | 0.559 |
| | | 720 | 0.454 | 0.432 | 0.680 | 0.450 | 0.419 | 0.506 | 0.677 | 0.429 | 0.712 | 0.589 |
| | Timer | 96 | 0.404 | 0.338 | 0.345 | 0.251 | 0.240 | 0.284 | 0.234 | **0.199** | 0.202 | 0.405 |
| | | 192 | 0.421 | 0.388 | 0.370 | 0.296 | 0.256 | 0.292 | 0.249 | **0.244** | 0.295 | 0.439 |
| | | 336 | 0.433 | 0.407 | 0.391 | 0.332 | 0.274 | 0.300 | 0.262 | 0.286 | 0.404 | **0.464** |
| | | 720 | 0.464 | 0.436 | 0.424 | 0.390 | 0.311 | 0.318 | 0.278 | 0.342 | 0.685 | 0.499 |
| | UniTS | 96 | 0.405 | 0.366 | 0.442 | 0.288 | 0.239 | 0.279 | 0.276 | 0.213 | 0.241 | 0.438 |
| | | 192 | 0.420 | 0.404 | 0.473 | 0.329 | 0.255 | 0.285 | 0.292 | 0.259 | 0.343 | 0.466 |
| | | 336 | 0.435 | 0.407 | 0.510 | 0.371 | **0.269** | 0.292 | 0.301 | 0.298 | 0.447 | 0.493 |
| | | 720 | 0.464 | 0.466 | 0.534 | 0.423 | 0.302 | 0.319 | 0.307 | 0.352 | 0.733 | 0.550 |
| | TTM | 96 | **0.393** | 0.342 | 0.368 | 0.254 | 0.252 | 0.326 | 0.253 | 0.205 | 0.201 | **0.396** |
| | | 192 | **0.411** | 0.381 | 0.392 | 0.294 | 0.266 | 0.334 | 0.259 | 0.247 | **0.294** | **0.436** |
| | | 336 | 0.425 | 0.408 | 0.721 | 0.414 | 0.326 | 0.468 | 0.966 | 0.344 | **0.399** | 0.476 |
| | | 720 | **0.443** | 0.436 | 0.716 | 0.458 | **0.189** | 0.481 | 0.933 | 0.383 | **0.658** | 0.511 |
| | Moment | 96 | 0.442 | 0.377 | 0.370 | 0.276 | 0.303 | 0.472 | 0.457 | 0.239 | 0.259 | 0.474 |
| | | 192 | 0.453 | 0.400 | 0.384 | 0.306 | 0.328 | 0.456 | 0.464 | 0.270 | 0.345 | 0.462 |
| | | 336 | 0.463 | 0.412 | 0.397 | 0.336 | 0.362 | 0.458 | 0.466 | 0.300 | 0.448 | 0.493 |
| | | 720 | 0.512 | 0.455 | 0.423 | 0.388 | 0.364 | 0.472 | 0.450 | 0.344 | 0.735 | 0.508 |
| | MOIRAI | 96 | 0.412 | 0.350 | 0.405 | 0.308 | 0.304 | 0.325 | 0.965 | 0.266 | 0.213 | 0.507 |
| | | 192 | 0.439 | 0.397 | 0.424 | 0.359 | 0.311 | 0.338 | 0.942 | 0.299 | 0.314 | 0.582 |
| | | 336 | 0.461 | 0.421 | 0.445 | 0.411 | 0.322 | - | 0.882 | 0.331 | 0.445 | 0.641 |
| | | 720 | 0.475 | 0.461 | 0.487 | 0.484 | 0.348 | - | 0.797 | 0.372 | 0.717 | 0.735 |
| | ROSE | 96 | 0.396 | **0.332** | **0.339** | **0.249** | **0.234** | **0.258** | **0.232** | 0.205 | 0.266 | 0.412 |
| | | 192 | 0.412 | **0.371** | **0.363** | **0.285** | 0.259 | **0.277** | **0.204** | 0.246 | 0.400 | 0.459 |
| | | 336 | **0.422** | 0.398 | **0.384** | **0.322** | 0.277 | **0.280** | 0.258 | 0.292 | 0.507 | 0.484 |
| | | 720 | 0.447 | **0.418** | 0.414 | **0.377** | 0.312 | **0.300** | **0.252** | 0.351 | 0.900 | 0.600 |
| LLM | GPT4TS | 96 | 0.445 | 0.377 | 0.380 | 0.279 | 0.292 | 0.288 | 0.305 | 0.244 | 0.246 | 0.486 |
| | | 192 | 0.455 | 0.402 | 0.394 | 0.311 | 0.306 | 0.292 | 0.335 | 0.274 | 0.340 | 0.505 |
| | | 336 | 0.467 | 0.421 | 0.406 | 0.342 | 0.321 | 0.295 | 0.343 | 0.304 | 0.449 | 0.515 |
| | | 720 | 0.511 | 0.458 | 0.434 | 0.394 | 0.345 | 0.315 | 0.350 | 0.346 | 0.739 | 0.532 |
| | S2IPLLM | 96 | 0.553 | 0.415 | 0.361 | 0.300 | - | - | 0.287 | 0.228 | 0.241 | 0.592 |
| | | 192 | 0.562 | 0.424 | 0.412 | 0.336 | - | - | 0.325 | 0.270 | 0.334 | 0.604 |
| | | 336 | 0.582 | 0.435 | 0.419 | 0.371 | - | - | 0.356 | 0.299 | 0.439 | 0.642 |
| | | 720 | 0.616 | 0.472 | 0.462 | 0.417 | - | - | 0.351 | 0.346 | 0.743 | 0.721 |
| | TimeLLM | 96 | 0.494 | 0.381 | 0.362 | 0.272 | - | - | - | 0.220 | 0.247 | 0.527 |
| | | 192 | 0.565 | 0.442 | 0.383 | 0.305 | - | - | - | 0.252 | 0.305 | 0.479 |
| | | 336 | 0.595 | 0.431 | 0.400 | 0.335 | - | - | - | 0.287 | 0.453 | 0.564 |
| | | 720 | 0.612 | 0.477 | 0.429 | 0.393 | - | - | - | 0.339 | 0.742 | 0.577 |
| | UniTime | 96 | 0.575 | 0.390 | 0.384 | 0.275 | 0.282 | 0.436 | 0.281 | 0.239 | 0.240 | 0.603 |
| | | 192 | 0.614 | 0.446 | 0.406 | 0.308 | 0.296 | 0.411 | 0.270 | 0.304 | 0.455 | 0.661 |
| | | 336 | 0.627 | 0.455 | 0.421 | 0.339 | 0.308 | 0.425 | 0.284 | 0.345 | 0.446 | 0.691 |
| | | 720 | 0.654 | 0.486 | 0.446 | 0.390 | 0.338 | 0.448 | 0.278 | 0.391 | 0.761 | 0.724 |
| Specific Model | PatchTST | 96 | 0.409 | 0.394 | 0.356 | 0.281 | 0.267 | 0.298 | 0.321 | 0.222 | 0.219 | 0.463 |
| | | 192 | 0.440 | 0.415 | 0.375 | 0.313 | 0.267 | 0.305 | 0.329 | 0.261 | 0.311 | 0.474 |
| | | 336 | 0.451 | 0.425 | 0.407 | 0.342 | 0.283 | 0.307 | 0.333 | 0.295 | 0.409 | 0.493 |
| | | 720 | 0.540 | 0.461 | 0.425 | 0.396 | 0.310 | 0.325 | 0.333 | 0.341 | 0.728 | 0.510 |
| | DLinear | 96 | 0.410 | 0.426 | 0.379 | 0.282 | 0.241 | 0.292 | 0.288 | 0.250 | 0.226 | 0.423 |
| | | 192 | 0.435 | 0.469 | 0.394 | 0.355 | **0.254** | 0.298 | 0.307 | 0.286 | 0.348 | 0.453 |
| | | 336 | 0.449 | 0.513 | 0.404 | 0.384 | 0.271 | 0.306 | 0.321 | 0.318 | 0.431 | 0.473 |
| | | 720 | 0.545 | 0.580 | 0.431 | 0.398 | 0.304 | 0.324 | 0.323 | 0.364 | 0.679 | **0.494** |
| | FITS | 96 | 0.396 | 0.345 | 0.345 | 0.254 | 0.249 | 0.290 | 0.255 | 0.225 | **0.199** | 0.422 |
| | | 192 | 0.418 | 0.379 | 0.365 | 0.291 | 0.260 | 0.294 | 0.267 | 0.261 | 0.295 | 0.455 |
| | | 336 | 0.435 | **0.396** | **0.384** | 0.326 | 0.279 | 0.308 | 0.273 | 0.295 | 0.406 | 0.477 |
| | | 720 | 0.458 | 0.425 | **0.413** | 0.381 | 0.313 | 0.347 | 0.277 | 0.341 | 0.684 | 0.501 |
| | iTransformer | 96 | 0.481 | 0.377 | 0.371 | 0.288 | 0.331 | 0.323 | 0.289 | 0.241 | 0.255 | 0.544 |
| | | 192 | 0.504 | 0.426 | 0.398 | 0.325 | 0.355 | 0.338 | 0.334 | 0.281 | 0.345 | 0.586 |
| | | 336 | 0.569 | 0.454 | 0.447 | 0.358 | 0.285 | 0.353 | 0.319 | 0.326 | 0.449 | 0.617 |
| | | 720 | 0.529 | 0.481 | 0.480 | 0.408 | 0.323 | 0.382 | 0.346 | 0.369 | 0.737 | 0.654 |
| | FEDformer | 96 | 0.475 | 0.443 | 0.493 | 0.334 | 0.374 | 0.475 | 0.693 | 0.316 | 0.304 | 0.480 |
| | | 192 | 0.505 | 0.499 | 0.549 | 0.371 | 0.388 | 0.546 | 0.578 | 0.327 | 0.581 | 0.531 |
| | | 336 | 0.535 | 0.452 | 0.576 | 0.514 | 0.405 | 0.577 | 0.638 | 0.393 | 0.666 | 0.590 |
| | | 720 | 0.559 | 0.584 | 0.595 | 0.594 | 0.425 | 0.462 | 0.630 | 0.564 | 0.825 | 0.739 |
| | TimesNet | 96 | 0.615 | 0.405 | 0.421 | 0.281 | 0.326 | 0.424 | 0.347 | 0.249 | 0.268 | 0.522 |
| | | 192 | 0.608 | 0.452 | 0.433 | 0.353 | 0.333 | 0.466 | 0.359 | 0.285 | 0.364 | 0.486 |
| | | 336 | 0.644 | 0.466 | 0.480 | 0.360 | 0.346 | 0.467 | 0.380 | 0.319 | 0.461 | 0.541 |
| | | 720 | 0.673 | 0.490 | 0.504 | 0.443 | 0.371 | 0.574 | 0.366 | 0.377 | 0.752 | 0.554 |
| | TimeMixer | 96 | 0.458 | 0.364 | 0.362 | 0.274 | 0.276 | 0.307 | 0.264 | 0.217 | 0.210 | 0.417 |
| | | 192 | 0.599 | 0.394 | 0.391 | 0.319 | 0.284 | 0.355 | 0.279 | 0.257 | 0.302 | 0.456 |
| | | 336 | 0.601 | 0.426 | 0.402 | 0.343 | 0.317 | 0.330 | 0.297 | **0.282** | 0.405 | 0.479 |
| | | 720 | 0.546 | 0.464 | 0.438 | 0.393 | 0.331 | 0.367 | 0.292 | **0.337** | 0.708 | 0.511 |

The maximum training duration was constrained to a maximum of 5 hours. Models that exceeded this threshold are represented with –.

## D.3 FULL-SHOT RESULTS

Table 15: Model results in the **full-shot setting**. The results are **MSE** of each prediction length.

| | Model | Horizon | ETTh1 | Weather | Exchange | ZafNoo | ILI | NASDAQ |
|---|---|---|---|---|---|---|---|---|
| TS Pretrain Model | Timer | 96 | 0.416 | 0.164 | 0.104 | 0.470 | 9.554 | 0.815 |
| | | 192 | 0.557 | 0.243 | 0.221 | 0.548 | 5.203 | 1.221 |
| | | 336 | 0.502 | 0.321 | 0.382 | 0.588 | 2.325 | 1.441 |
| | | 720 | 0.525 | 0.349 | 0.965 | 0.637 | 6.151 | 1.517 |
| | UniTS | 96 | 0.399 | 0.147 | 0.444 | 0.444 | 2.983 | 0.792 |
| | | 192 | 0.441 | 0.191 | 0.507 | 0.507 | 3.119 | 1.094 |
| | | 336 | 0.503 | **0.243** | 0.489 | 0.563 | 2.765 | 1.452 |
| | | 720 | 0.468 | 0.317 | 0.997 | 0.602 | 2.513 | 1.364 |
| | TTM | 96 | **0.359** | **0.146** | **0.082** | **0.421** | 4.313 | 1.047 |
| | | 192 | **0.389** | **0.190** | **0.173** | **0.482** | 4.631 | 1.329 |
| | | 336 | **0.418** | 0.292 | 0.317 | 0.543 | 4.422 | 1.544 |
| | | 720 | **0.422** | 0.359 | 0.824 | 0.604 | 4.268 | 1.610 |
| LLM | GPT4TS | 96 | 0.373 | 0.154 | 0.117 | 0.443 | 3.627 | 1.003 |
| | | 192 | 0.421 | 0.200 | 0.232 | 0.499 | 3.625 | 1.219 |
| | | 336 | 0.428 | 0.251 | 0.463 | 0.536 | 3.833 | 1.281 |
| | | 720 | 0.459 | 0.316 | 1.086 | 0.579 | 3.972 | **1.206** |
| | UniTime | 96 | 0.392 | 0.153 | 0.105 | 0.448 | 3.835 | 0.772 |
| | | 192 | 0.487 | 0.197 | 0.221 | 0.503 | 3.917 | 1.081 |
| | | 336 | 0.504 | 0.250 | 0.403 | 0.543 | 2.691 | 1.278 |
| | | 720 | 0.617 | 0.319 | 1.018 | 0.585 | 2.752 | 1.264 |
| Specific Model | PatchTST | 96 | 0.376 | 0.149 | 0.083 | 0.444 | 1.840 | 0.649 |
| | | 192 | 0.399 | 0.193 | 0.176 | 0.498 | **1.724** | 0.821 |
| | | 336 | 0.418 | 0.244 | **0.301** | 0.530 | 1.762 | 1.169 |
| | | 720 | 0.450 | **0.314** | 0.847 | 0.574 | **1.752** | 1.268 |
| | Dlinear | 96 | 0.371 | 0.170 | **0.082** | 0.434 | 2.208 | 0.830 |
| | | 192 | 0.404 | 0.212 | 0.186 | 0.484 | 2.032 | 1.356 |
| | | 336 | 0.434 | 0.257 | 0.328 | **0.518** | 2.209 | 1.817 |
| | | 720 | 0.469 | 0.318 | **0.801** | 0.548 | 2.292 | 2.011 |
| | FITS | 96 | 0.376 | 0.172 | **0.082** | 0.454 | 2.182 | 0.709 |
| | | 192 | 0.400 | 0.215 | **0.173** | 0.509 | 2.330 | 1.058 |
| | | 336 | 0.419 | 0.261 | 0.317 | 0.550 | 2.761 | 1.255 |
| | | 720 | 0.435 | 0.326 | 0.825 | 0.593 | 2.929 | 1.258 |
| | iTransformer | 96 | 0.386 | 0.159 | 0.086 | 0.462 | **1.783** | 0.570 |
| | | 192 | 0.424 | 0.200 | 0.177 | 0.517 | 1.746 | **0.769** |
| | | 336 | 0.449 | 0.253 | 0.331 | 0.556 | **1.716** | 1.188 |
| | | 720 | 0.495 | 0.321 | 0.846 | 0.606 | 1.960 | 1.422 |
| | FedFormer | 96 | 0.379 | 0.223 | 0.136 | 0.475 | 2.400 | 0.627 |
| | | 192 | 0.419 | 0.252 | 0.239 | 0.544 | 2.410 | 0.885 |
| | | 336 | 0.455 | 0.327 | 0.438 | 0.595 | 2.592 | **1.139** |
| | | 720 | 0.474 | 0.424 | 1.117 | 0.697 | 2.539 | 1.251 |
| | TimesNet | 96 | 0.389 | 0.170 | 0.109 | 0.479 | 2.009 | **0.563** |
| | | 192 | 0.440 | 0.222 | 0.213 | 0.491 | 2.552 | 0.905 |
| | | 336 | 0.482 | 0.293 | 0.358 | 0.551 | 1.956 | 1.218 |
| | | 720 | 0.525 | 0.360 | 1.004 | 0.627 | 2.178 | 1.298 |
| | TimeMixer | 96 | 0.373 | 0.147 | 0.084 | 0.441 | 1.807 | 0.720 |
| | | 192 | 0.415 | 0.192 | 0.178 | 0.498 | 1.896 | 0.951 |
| | | 336 | 0.454 | 0.247 | 0.386 | 0.545 | 1.753 | 1.214 |
| | | 720 | 0.501 | 0.318 | 0.884 | 0.587 | 1.828 | 1.682 |

Table 16: Model results in the **full-shot setting**. The results are **MAE** of each prediction length.

| | Model | Horizon | ETTh1 | Weather | Exchange | ZafNoo | ILI | NASDAQ |
|---|---|---|---|---|---|---|---|---|
| TS Pretrain Model | Timer | 96 | 0.423 | 0.210 | 0.228 | 0.416 | 2.391 | 0.715 |
| | | 192 | 0.489 | 0.287 | 0.336 | 0.456 | 1.638 | 0.784 |
| | | 336 | 0.480 | 0.338 | 0.448 | 0.479 | 1.027 | 0.842 |
| | | 720 | 0.500 | 0.346 | 0.730 | 0.505 | 1.897 | 0.856 |
| | UniTS | 96 | 0.421 | 0.203 | 0.411 | 0.411 | 1.235 | 0.674 |
| | | 192 | 0.445 | 0.244 | 0.451 | 0.451 | 1.260 | 0.783 |
| | | 336 | 0.493 | 0.286 | 0.522 | 0.479 | 1.181 | 0.874 |
| | | 720 | 0.484 | 0.337 | 0.756 | 0.498 | 1.106 | 0.848 |
| | TTM | 96 | **0.390** | **0.195** | 0.200 | **0.389** | 1.512 | 0.806 |
| | | 192 | **0.408** | **0.237** | 0.297 | **0.428** | 1.559 | 0.885 |
| | | 336 | **0.422** | 0.333 | 0.408 | 0.464 | 1.506 | 0.924 |
| | | 720 | **0.439** | 0.373 | 0.682 | 0.497 | 1.468 | 0.925 |
| LLM | GPT4TS | 96 | 0.397 | 0.205 | 0.245 | 0.414 | 1.365 | 0.784 |
| | | 192 | 0.428 | 0.246 | 0.348 | 0.448 | 1.354 | 0.844 |
| | | 336 | 0.437 | 0.287 | 0.508 | 0.468 | 1.455 | 0.855 |
| | | 720 | 0.483 | 0.333 | 0.783 | 0.498 | 1.472 | 0.828 |
| | UniTime | 96 | 0.413 | 0.205 | 0.231 | 0.419 | 1.433 | 0.661 |
| | | 192 | 0.478 | 0.244 | 0.344 | 0.452 | 1.467 | 0.769 |
| | | 336 | 0.489 | 0.285 | 0.466 | 0.475 | 1.134 | 0.835 |
| | | 720 | 0.552 | 0.334 | 0.762 | 0.498 | 1.150 | 0.842 |
| Specific Model | PatchTST | 96 | 0.396 | 0.196 | 0.200 | 0.426 | 0.835 | 0.567 |
| | | 192 | 0.416 | 0.240 | **0.298** | 0.456 | **0.845** | 0.682 |
| | | 336 | 0.432 | **0.281** | 0.397 | 0.480 | **0.863** | 0.793 |
| | | 720 | 0.469 | 0.332 | 0.693 | 0.499 | **0.894** | 0.828 |
| | Dlinear | 96 | 0.392 | 0.230 | 0.204 | 0.411 | 1.031 | 0.666 |
| | | 192 | 0.413 | 0.267 | 0.325 | 0.444 | 0.981 | 0.862 |
| | | 336 | 0.435 | 0.305 | 0.435 | **0.464** | 1.063 | 0.990 |
| | | 720 | 0.489 | **0.356** | **0.679** | **0.486** | 1.086 | **0.104** |
| | FITS | 96 | 0.396 | 0.225 | **0.199** | 0.442 | 1.002 | 0.645 |
| | | 192 | 0.418 | 0.261 | 0.295 | 0.455 | 1.051 | 0.778 |
| | | 336 | 0.435 | 0.295 | 0.406 | 0.477 | 1.184 | 0.834 |
| | | 720 | 0.458 | 0.341 | 0.684 | 0.501 | 1.127 | 0.833 |
| | iTransformer | 96 | 0.405 | 0.208 | 0.208 | 0.431 | 0.846 | **0.540** |
| | | 192 | 0.440 | 0.248 | 0.299 | 0.464 | 0.860 | **0.632** |
| | | 336 | 0.460 | 0.289 | 0.417 | 0.486 | 0.898 | **0.773** |
| | | 720 | 0.487 | 0.338 | 0.693 | 0.509 | 0.977 | 0.846 |
| | FedFormer | 96 | 0.419 | 0.292 | 0.267 | 0.441 | 1.020 | 0.547 |
| | | 192 | 0.443 | 0.322 | 0.353 | 0.476 | 1.005 | 0.659 |
| | | 336 | 0.464 | 0.371 | 0.486 | 0.521 | 1.033 | 0.786 |
| | | 720 | 0.488 | 0.419 | 0.811 | 0.543 | 1.070 | 0.783 |
| | TimesNet | 96 | 0.412 | 0.219 | 0.238 | 0.424 | 0.926 | 0.563 |
| | | 192 | 0.443 | 0.264 | 0.163 | 0.446 | 0.997 | 0.687 |
| | | 336 | 0.465 | 0.310 | 0.298 | 0.479 | 0.919 | 0.783 |
| | | 720 | 0.483 | 0.355 | 0.797 | 0.511 | 0.962 | 0.781 |
| | TimeMixer | 96 | 0.401 | 0.198 | 0.207 | 0.396 | **0.820** | 0.612 |
| | | 192 | 0.425 | 0.243 | 0.300 | 0.444 | 0.927 | 0.699 |
| | | 336 | 0.453 | 0.284 | 0.450 | 0.479 | 0.866 | 0.795 |
| | | 720 | 0.481 | 0.330 | 0.707 | 0.498 | 0.930 | 0.884 |

