# OpenReview forum: "FoundTS: Comprehensive and Unified Benchmarking of Foundation Models for Time Series Forecasting"
_ICLR.cc/2025/Conference — ICLR 2025 Conference Withdrawn Submission_

### Official Review · Reviewer_ZGbi · 2024-11-01

**Soundness:** 2
**Presentation:** 3
**Contribution:** 1
**Rating:** 5
**Confidence:** 4

**Summary:**

This paper proposes to evaluate different families of time-series forecasting models in a unified framework.
The benchmark consists of several well-known datasets which are grouped based on their origin and dataset characteristics such as seasonality.
The authors proceed to evaluate a variety of models (foundation models as well as customly trained approaches) in different evaluation settings such as zero-shot.

**Strengths:**

- The first benchmark to compare time-series foundation models.
- The benchmark includes a variety of datasets and methods but also importantly evaluation settings like zero-shot and few-shot which have become more relevant for foundation models.
- Interesting finding that LLM-based time-series foundation models work worse than time-series specific foundation models.

**Weaknesses:**

- Since the datasets used in the benchmark are publicly available and commonly used in time-series forecasting, the foundation models considered in this benchmark may be contaimnated. Could you please provide a list of datasets each foundation model was trained and compare it to the datasets used in your benchmark?
- Several foundation models like Chronos or ForecastPFN were at least partially trained using synthetic data. In reference to weakness 1, I believe that a fair evaluation of foundation models should also include some synthetic data showing some realistic data characteristics. Could you please consider adding maybe the data from the ForecastPFN or Chronos data generator to your benchmark and report results on them?
- Could you explain your selection criteria for the foundation models you evaluated? Some foundation models like Chronos and ForecastPFN are missing and I think they would improve the submission as forecastpfn is only trained on synthetic data and Chronos is widely used at the moment.


Minor:
- Please check the references and make sure to use citet and citep where appropriate. I noticed this especially in section 3.2.1.
- The paper is slightly above page limit with the reproducibility section going to page 11.

**Questions:**

-

---

### Official Review · Reviewer_p1Rf · 2024-11-02

**Soundness:** 2
**Presentation:** 3
**Contribution:** 2
**Rating:** 5
**Confidence:** 3

**Summary:**

This paper presents FoundTS, a benchmark for fair and comprehensive evaluation of Time Series Forecasting foundation models. FoundTS includes diverse models—LLM-based, time series pre-trained, and specific models—and supports multiple evaluation strategies (zero-shot, few-shot, and full-shot) in a standardized pipeline. Key results show time series pre-trained models excel in few-shot tasks, while LLM-based models struggle, indicating a need for optimization. Smaller models like ROSE and TTM offer a good performance-efficiency balance, and no single model dominates across scenarios, highlighting the need for versatile TSF models.

**Strengths:**

1. This work incorporates a comprehensive range of models and settings, including LLM-based and foundational time series models, and evaluates them across zero-shot, few-shot, and full-shot scenarios.

2. It includes a wide variety of data domains, enhancing the benchmark’s relevance and robustness across different applications.

3. The insights from this study are clear and may possibly benefit for industry professionals.

**Weaknesses:**

1. The takeaway conclusions are not impressive, such as 'Foundational models are more powerful than specific models'; 'Foundational TS models are better than LLM-based models'; and 'Scaling law doesn't hold for TS models'. Most of them have been noted in prior studies [1,2,3]. Similarly, the suggestions for improving foundational TS models lack specificity and fail to offer new insights.

[1] MOMENT: A Family of Open Time-series Foundation Models. ICML 2024

[2] A Decoder-Only Foundation Model for Time-Series Forecasting. ICML 2024

[3] Are Language Models Actually Useful for Time Series Forecasting? ​​https://arxiv.org/pdf/2406.16964v1​

2. Figures 3 and 5 are difficult to interpret; additional context and explanation are needed to clarify the insights these figures provide.

3. Beyond simply ranking TS models, it would be beneficial to highlight actionable guidance or overarching principles in TS forecasting that could directly benefit industry professionals using FoundTS.

4. Section 4.2.4’s conclusions (points 2 and 3) are unconvincing. As illustrated in Figure 1, LLM-based models take prompts during pretraining, but they cannot accurately interpret prompts if pre-trained weights are not effectively loaded.

5. The open-sourced code provided is currently inaccessible.

**Questions:**

Please refer to the weakness.

---

### Official Review · Reviewer_2Ngw · 2024-11-04

**Soundness:** 2
**Presentation:** 2
**Contribution:** 2
**Rating:** 3
**Confidence:** 4

**Summary:**

This paper presents FoundTS, a benchmark for time series foundation models. The key motivation is that the previous work in this domain has applied different hyperparameter settings. So it is not very clear whether the evaluation was fair. In this paper, the authors build a standardized pipeline to evaluate time series forecasting tasks in few-shot and full-shot settings.

**Strengths:**

1. This paper benchmarks time series foundation models.
2. The presentation is clear.
3. The benchmark additionally presents some interesting analytical experiments.

**Weaknesses:**

1. A foundation model is expected to be able to be applied to various downstream tasks. However, the paper only benchmarks time series forecasting, while ignoring other tasks such as anomaly detection, classification, clustering, etc.
2. The main results (Tables 4, 5, and 6) do not have much new compared with previous work. Similar results have already been reported in the previous papers.
3. The standard deviation is not reported in all the results.
4. Some related work in benchmarking or evaluating time series foundation models are not discussed, such as [1] [2] [3] [4]

[1] GIFT-Eval: A Benchmark For General Time Series Forecasting Model Evaluation
[2] Evaluating Large Language Models on Time Series Feature Understanding: A Comprehensive Taxonomy and Benchmark
[3] Understanding Different Design Choices in Training Large Time Series Models
[4] TSGBench: Time Series Generation Benchmark

**Questions:**

See weakness.

---

### Official Review · Reviewer_BqMP · 2024-11-07

**Soundness:** 2
**Presentation:** 1
**Contribution:** 1
**Rating:** 3
**Confidence:** 4

**Summary:**

The paper presents results of several time series models on a set of datasets in several settings - zero-shot, few-shot, and full-shot. Some analysis was performed on characteristics of the models such as "channel independence" and "channel dependence".

**Strengths:**

The paper presents a good effort on the important task of benchmarking and quantitatively assessing the recent advances in deep learning for time series forecasting.

**Weaknesses:**

Unfortunately, the paper does not quite cover sufficient scope to constitute a comprehensive benchmark, and also falls short of presenting some significant insight into the methods. The datasets are insufficiently diverse, for example, ETTh1/h2/m1/m2 are in fact from the same source and should be considered a single dataset instead. A single dataset for a domain is insufficient to present robust results, especially for a benchmark paper. There seems to be no consideration regarding the sampling frequency of the datasets, which is a critical characteristic of time series, and the analysis of the properties of characteristics of the datasets is not well presented - just a table in the appendix. The reader is not guided towards the significance of the differences in these characteristics.

There are also some concerns regarding the rigor of the experimental setup. For example, TimesFM has been pre-trained on the Electricity and Traffic datasets, and should not qualify for the zero-shot setting for these datasets. Furthermore, for a benchmark paper, there is a greater onus to ensure that each model is well tuned without bias - such details are missing from the exposition of the paper.

I would also recommend the authors to revisit terms such as "specific model", and use terms which are more aligned with existing literature.

**Questions:**

-

---

### Note · Authors · 2024-11-21

I have read and agree with the venue's withdrawal policy on behalf of myself and my co-authors.